# A SUMO E3 ligase promotes long non-coding RNA transcription to regulate small RNA-directed DNA elimination

Salman Shehzada[†], Tomoko Noto, Julie Saksouk, Kazufumi Mochizuki*

Institute of Human Genetics (IGH), CNRS, University of Montpellier, Montpellier, France

**Abstract** Small RNAs target their complementary chromatin regions for gene silencing through nascent long non-coding RNAs (lncRNAs). In the ciliated protozoan *Tetrahymena*, the interaction between Piwi-associated small RNAs (scnRNAs) and the nascent lncRNA transcripts from the somatic genome has been proposed to induce target-directed small RNA degradation (TDSD), and scnRNAs not targeted for TDSD later target the germline-limited sequences for programmed DNA elimination. In this study, we show that the SUMO E3 ligase Ema2 is required for the accumulation of lncRNAs from the somatic genome and thus for TDSD and completing DNA elimination to make viable sexual progeny. Ema2 interacts with the SUMO E2 conjugating enzyme Ubc9 and enhances SUMOylation of the transcription regulator Spt6. We further show that Ema2 promotes the association of Spt6 and RNA polymerase II with chromatin. These results suggest that Ema2-directed SUMOylation actively promotes lncRNA transcription, which is a prerequisite for communication between the genome and small RNAs.

## Editor's evaluation

This important study demonstrates that protein SUMOylation is essential for programmed DNA elimination guided by small RNAs during conjugation in *Tetrahymena* ciliates. The authors present convincing evidence that the E3 SUMO ligase Ema2 is necessary for the production of long non-coding RNAs from the somatic nucleus, targeted small RNA degradation, and DNA elimination. The authors also show that the transcription regulator Spt6 is a SUMOylation target of Ema2, though the relevance of this is not completely established. This paper is of broad significance and will appeal to those interested in non-coding RNA biology, the control of programmed genome rearrangements, or ciliate biology.

## Introduction

Small RNAs of approximately 20–30 nucleotides that are complexed with Argonaute family proteins target either mRNAs for post-transcriptional silencing or chromatin regions for transcriptional gene silencing (*Holoch and Moazed, 2015*; *Wilson and Doudna, 2013*). For the latter process, small RNAs are generally considered to recognize their genomic targets via nascent lncRNAs to induce heterochromatin formation. Therefore, chromatin regions that are targeted for silencing by small RNAs must be paradoxically transcribed to provide nascent lncRNAs.

In fission yeast, small interfering RNAs (siRNAs) mediate the deposition of histone 3 lysine 9 di- and trimethylation (H3K9me2/3) for heterochromatin assembly at centromeric repeats (*Hall et al., 2002*; *Volpe et al., 2002*). While the HP1 protein Swi6 binds to H3K9me2/3 for transcriptional silencing, phosphorylation of histone H3 serine 10 at the M phase of the cell cycle evicts Swi6, passively

*For correspondence:
kazufumi.mochizuki@cnrs.fr;
kazufumi.mochizuki@igh.cnrs.fr

Present address: †Department
of Genetic Medicine and
Development, University of
Geneva, Geneva, Switzerland

Competing interest: The authors
declare that no competing
interests exist.

Reviewing Editor: Adèle
L Marston, University of
Edinburgh, United Kingdom

allowing lncRNA transcription from the centromeric repeats at G1 and S phase and thus promoting H3K9me2/3 deposition to newly assembled nucleosomes (*Chen et al., 2008*; *Kloc et al., 2008*). In contrast, the HP1 paralog Rhino in fruit flies, which is specifically enriched at PIWI-interacting RNA (piRNA) clusters by binding to H3K9me2/3 (*Le Thomas et al., 2014*), actively recruits dedicated variants of basal transcription factors, allowing lncRNA transcription from heterochromatin while preventing mRNA transcription from transposons in the same loci (*Andersen et al., 2017*). A similar active heterochromatin-dependent lncRNA transcription has been reported in plants, where SHH1, a reader of H3K9me2/3 as well as mono-methylated H3K9, recruits the plant-specific RNA polymerase IV (*Law et al., 2013*; *Law et al., 2011*).

Because small RNA-producing loci are also small RNA targets in most of the studied small RNA-directed heterochromatin formation processes, it poses a challenge to separately investigate lncRNA transcription for small RNA biogenesis and that for small RNA-dependent recruitment of downstream effectors in these processes. In contrast, the source and target loci of small RNAs reside in different nuclei during programmed DNA elimination in some ciliated protozoans such as *Tetrahymena thermophila* and *Paramecium tetraurelia*, which provides a unique system to study the mechanism and roles of lncRNA transcription in small RNA-directed chromatin regulation.

In most ciliates, each cell contains two types of nuclei, the diploid germline micronucleus (MIC) and the polyploid somatic macronucleus (MAC). During conjugation, a sexual reproduction process in ciliates, the MIC undergoes meiosis and fertilization, followed by the formation of the new MIC and MAC, while the parental MAC is degraded (*Chalker et al., 2013*). Programmed DNA elimination occurs in the new MAC in most ciliates. In *Tetrahymena,* this process downsizes the 200 Mb MIC genome to the 103 Mb MAC genome by removing ~12,000 internal elimination sequences (IESs), many of which include transposons, followed by re-ligation of the remaining macronuclear-destined sequences (MDSs) forming the MAC chromosomes (*Noto and Mochizuki, 2018*).

DNA elimination in *Tetrahymena* and *Paramecium* is regulated by three types of lncRNAs, which occur in different nuclei at distinct times (*Aronica et al., 2008*; *Schoeberl and Mochizuki, 2011*). During meiotic prophase in *Tetrahymena*, lncRNAs are transcribed bidirectionally in the MIC in a genome-wide manner by RNA polymerase II and dedicated conjugation-specific Mediator-associated proteins (*Chalker and Yao, 2001*; *Garg et al., 2019*; *Mochizuki and Gorovsky, 2004a*; *Schoeberl et al., 2012*; *Tian et al., 2019*). The sexual reproduction-specific Spt4 and Spt5 paralogs are specifically involved in the MIC transcription in *Paramecium* (*Gruchota et al., 2017*; *Owsian et al., 2022*), while *Tetrahymena* genome does not encode such specialized Spt4/Spt5 paralogs. These micronuclear long non-coding RNAs (MIC-lncRNAs) are processed into small RNAs (~29 nt in *Tetrahymena* and 25-nt in *Paramecium*), called scnRNAs, by Dicer homologs (*Lepère et al., 2009*; *Malone et al., 2005*; *Mochizuki and Gorovsky, 2005*) and loaded into Piwi-clade Argonaute proteins (*Bouhouche et al., 2011*; *Mochizuki et al., 2002*; *Noto et al., 2010*).

The Piwi-scnRNA complex then moves into the parental MAC, where parental macronuclear long non-coding RNAs (pMAC-lncRNAs) are transcribed bidirectionally during the mid-conjugation stages (*Woo et al., 2016*). The RNA helicase Ema1 in *Tetrahymena* promotes the interaction between pMAC-lncRNAs and scnRNAs (*Aronica et al., 2008*). Reminiscently of target-directed micro RNA degradation (TDMD) (*Han and Mendell, 2023*), this interaction induces TDSD, leading to the selective retention of IES-derived scnRNAs (*Aronica et al., 2008*; *Mochizuki and Gorovsky, 2004b*; *Noto and Mochizuki, 2018*; *Noto et al., 2015*; *Schoeberl et al., 2012*). A similar TDSD has also been suggested in *Paramecium,* and the importance of pMAC-lncRNAs for DNA elimination was demonstrated by disrupting certain pMAC-lncRNAs by RNAi in this ciliate (*Lepère et al., 2008*). Although mRNAs are transcribed in the parental MAC, it remains unclear whether they also can induce TDSD and how mRNAs and pMAC-lncRNAs can be transcribed from overlapping locations. Also, although trimethylation of histone 3 lysine 27 (H3K27me3) occurs in the parental MAC in a scnRNA- and Polycomb repressive complex 2 (PRC2)-dependent manner in both *Tetrahymena* and *Paramecium* (*Lhuillier-Akakpo et al., 2014*; *Liu et al., 2007*), the role of H3K27me3 in the parental MAC, if any, is unclear.

When the new MAC develops, the retained Piwi-scnRNA complexes translocate to the new MAC, where yet another type of lncRNA, new macronuclear non-coding RNAs (nMAC-lncRNAs), is transcribed. The interaction between nMAC-lncRNA and scnRNA, which is also dependent on Ema1 (*Aronica et al., 2008*), is believed to recruit PRC2, which catalyzes both H3K9me2/3 and H3K27me3 for IES-specific heterochromatin assembly (*Frapporti et al., 2019*; *Liu et al., 2007*; *Miró-Pina et al.,*

*2022*; *Wang et al., 2022*; *Xu et al., 2021*), and facilitate the secondary production of scnRNAs from nMAC-lncRNAs, which further promotes heterochromatin assembly (*Allen et al., 2017*; *Noto et al., 2015*). Then, IESs are excised by domesticated PiggyBac transposases (*Baudry et al., 2009*; *Bischerour et al., 2018*; *Cheng et al., 2010*; *Vogt and Mochizuki, 2013*). While nMAC-lncRNA production occurs prior to the excision of IESs in *Tetrahymena* (*Mutazono et al., 2019*), it also occurs from excised IESs that are concatenated and circularized in *Paramecium* (*Allen et al., 2017*).

Among the above three lncRNAs, pMAC-lncRNA does not produce small RNAs and is believed to be specialized for receptor function. Therefore, pMAC-lncRNA transcription and the following TDSD in the parental MAC provide a unique paradigm to investigate how the genome is transcribed to communicate with small RNAs. In this study, we show that Ema2, the conjugation-specific E3 ligase for a small ubiquitin-like modifier (SUMO), is required for pMAC-lncRNA transcription in *Tetrahymena* and thus provides a tool to dissect the role of and the molecular mechanism for the transcription of this lncRNA.

## Results

### Ema2 is exclusively expressed during conjugation and localized in the MAC

As part of our systematic investigation into genes highly upregulated during conjugation (*Loidl, 2021*), we explored the function of *EMA2* (*TTHERM_00113330*). *EMA2* mRNA is exclusively expressed during conjugation (*Figure 1A*). The encoded Ema2 protein tagged with HA at the endogenous *EMA2* locus was not detectable in the vegetative cells (*Figure 1B*, Vg) and first appeared in the MAC during conjugation (*Figure 1B*, 3 hr post-induction of mating [hpm] and 6 hpm). At the onset of new MAC development, Ema2 disappeared from the parental MAC and appeared in the new MAC (*Figure 1B*, 8 hpm), which later faded away (*Figures 1B*, 12, and 14 hpm). The conjugation-specific expression and the localization switch from the parental to the new MAC are reminiscent of the factors involved in DNA elimination such as the Piwi protein Twi1, which is loaded by scnRNAs, and PRC2 (*Liu et al., 2007*; *Mochizuki et al., 2002*; *Noto et al., 2010*).

### Ema2 is required for completing DNA elimination

DNA elimination of exconjugants (sexual progeny) at 36 hpm was analyzed by DNA fluorescent in situ hybridization (FISH) using probes complementary to the transposable element Tlr1 (*Wuitschick et al., 2002*). DNA elimination is completed by ~14–18 hpm in wild-type cells (*Austerberry et al., 1984*; *Mutazono et al., 2019*), and the Tlr1 element was detected only in the MICs in the exconjugants from wild-type cells (*Figure 2A*, WT). In contrast, the Tlr1 element was detected in both the MICs and the MACs in the exconjugants from the *EMA2* somatic KO strains, in which all *EMA2* copies in the MAC were disrupted (*Shehzada and Mochizuki, 2022*; *Figure 2A*, KO). The intensity of the FISH signal in the new MACs was lower in the exconjugants from the *EMA2* KO cells than in those from the *TWI1* KO cells (*Figure 2B*), in the latter of which DNA elimination is known to be completely blocked (*Noto et al., 2015*). Therefore, DNA elimination was partially blocked in the exconjugants of *EMA2* KO cells. Consistent with the requirement of DNA elimination in the viability of sexual progeny (*Cheng et al., 2010*; *Vogt and Mochizuki, 2013*), *EMA2* KO cells did not produce viable progeny (*Figure 2C*). Altogether, we conclude that maternally expressed *EMA2* is required for completing DNA elimination.

### Ema2 is required for target-directed small RNA degradation (TDSD)

We next asked whether *EMA2* is involved in TDSD. The production of scnRNAs occurs from IESs and their surrounding MDS regions in the MIC at the early conjugation stages (~2–3 hpm), and scnRNAs complementary to the MAC genome (=MDSs) are subjected to TDSD in the parental MAC in the mid-conjugation stages (~3–7 hpm), resulting in the selective retention of IES-derived scnRNAs that later target IESs for DNA elimination in the new MAC (*Aronica et al., 2008*; *Mochizuki and Gorovsky, 2004b*; *Noto et al., 2015*; *Schoeberl et al., 2012*). We first compared scnRNAs at different conjugation stages by northern blot analysis using the 50-nt Mi-9 probe that is complementary to a repetitive sequence found in both IESs and MDSs (*Aronica et al., 2008*). Because scnRNAs detected by this probe are complementary to the MAC genome, in wild-type cells, the Mi-9-complementary scnRNAs were detected at 3 hpm, reduced at 4.5 hpm, and became undetectable at 6 hpm due to TDSD

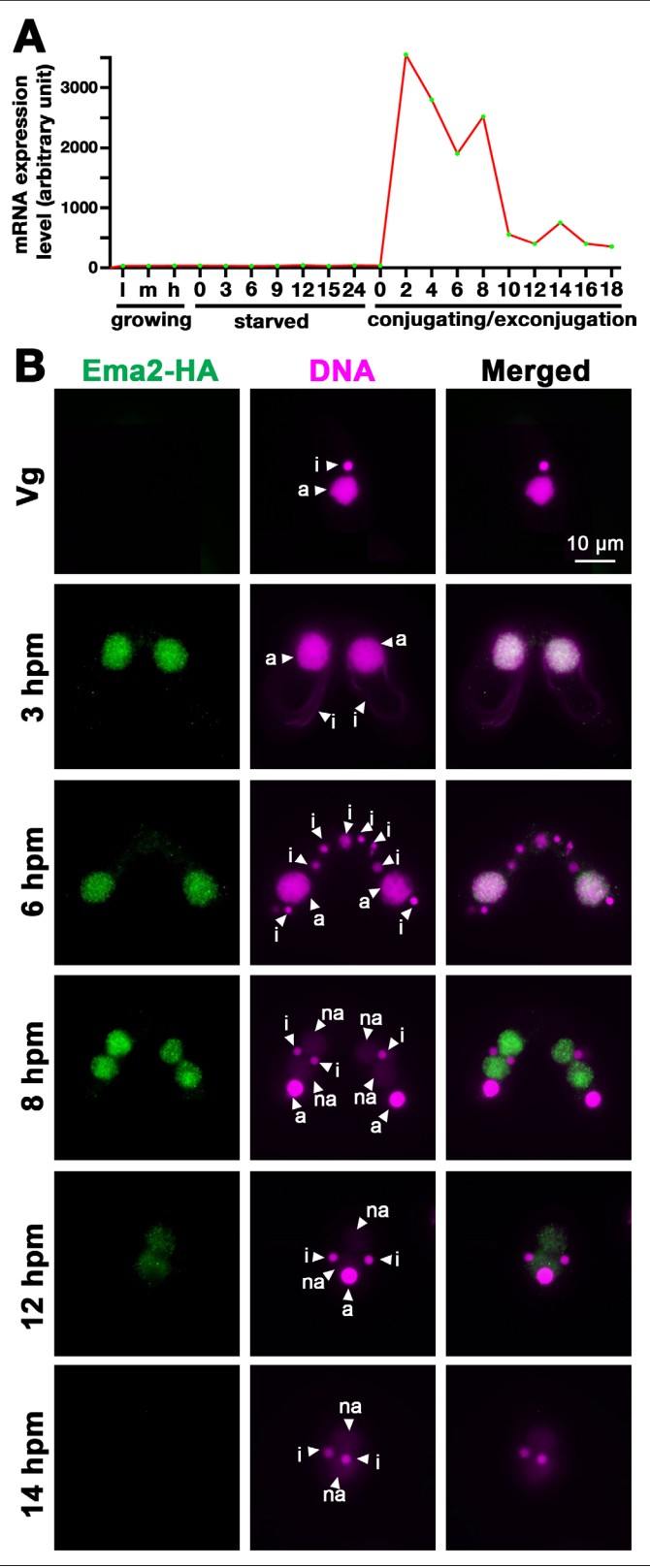

**Figure 1.** Ema2 is expressed during conjugation and localized in the macronucleus (MAC). (**A**) *EMA2* mRNA expression levels (in an arbitrary unit) in growing *Tetrahymena* cells in low (l), middle (m), and high (h) cell concentrations, starved cells from 0 to 24 hr, and cells in the conjugation and post conjugation stages from 0 to 18 hr post-mixing (hpm) are shown. The mRNA expression data were obtained from *Miao et al., 2009*. (**B**) Ema2

*Figure 1 continued on next page*

*Figure 1 continued*
localization. Two Ema2-HA strains were mated and fixed at the indicated time points (Vg = vegetative cell). An anti-HA antibody was used to localize Ema2-HA (green), and DNA was stained with DAPI (magenta). The micronucleus (MIC), the parental MAC, and the newly formed MAC are marked with arrowheads with 'i,' 'a,' and 'na,' respectively. All pictures share the scale bar.

(*Figure 3A*, WT). In contrast, the Mi-9-complementary scnRNAs remained detectable at 6 hpm in the *EMA2* KO cells (*Figure 3A*, KO), suggesting that Ema2 is required for TDSD.

We also analyzed TDSD by small RNA sequencing at 3 and 8 hpm. Sequenced 26- to 32-nt small RNAs, which correspond to scnRNAs (*Schoeberl et al., 2012*), were mapped to the genomic tiles of MDSs and IESs (see Materials and methods). Although scnRNAs that mapped many of the MDS tiles were greatly reduced from 3 to 8 hpm due to TDSD in the wild-type cells (*Figure 3B*, left, WT), those in the *EMA2* KO cells were only slightly reduced by 8 hpm (*Figure 3B*, left, *EMA2* KO). Because TDSD takes place concurrently with the scnRNA production (*Schoeberl et al., 2012*), the increased abundance of MDS-complementary scnRNAs at 3 hpm in the *EMA2* KO cells compared to the wild-type cells can also be attributed to the necessity of Ema2 in TDSD. This TDSD defect in the *EMA2* KO cells seems milder than that in the *EMA1* KO cells, in which the amount of MDS-complementary scnRNAs remained constant until 8 hpm (*Figure 3B*, left, *EMA1* KO), indicating that Ema1 and Ema2 act differently in TDSD. In contrast, the amounts of scnRNAs complementary to IES tiles at 3 hpm and 8 hpm were comparable in all strains (*Figure 3B*, right), which is consistent with previous observations that only scnRNAs complementary to MDSs are targeted for TDSD. These results suggest that Ema2 is required for TDSD at the genome-wide level. Because it is known that loss of TDSD results in a partial block of DNA elimination (*Aronica et al., 2008*), the mild DNA elimination defect in *EMA2* KO cells (*Figure 2B*) can be explained by the requirement of Ema2 in TDSD.

## Ema2 is required for the accumulation of SUMOylated proteins during conjugation

Ema2 possesses an SP-RING domain that has been found in many SUMO E3 ligases (*Hochstrasser, 2001*). The SP-RING domain of Ema2 is atypical (*Figure 4A*) in that the first cysteine of the zinc ion-binding residues in typical counterparts is replaced by histidine and some of the stabilizer residues (*Duan et al., 2009*; *Yunus and Lima, 2009*) are likely missing. SUMOylation is catalyzed by the sequential actions of E1, E2, and in most cases, E3 enzymes and thus Ema2 should interact with the E2 enzyme if it acts in SUMOylation. We indeed found that Ema2 can directly interact with the *Tetrahymena* SUMO E2 enzyme Ubc9 in vitro (*Figure 4B*), suggesting that Ema2 is a bona fide SUMO E3 ligase.

To examine the role of Ema2 in SUMOylation, we expressed HA-tagged Smt3 (HA-Smt3) in an *EMA2* KO strain. SUMO is solely encoded by *SMT3* in *Tetrahymena* (*Nasir et al., 2015*), and HA-Smt3 could replace the essential function of endogenous Smt3 (*Figure 4—figure supplement 1*). We then crossed this strain with either a wild-type strain (called the WT cross) or another *EMA2* KO strain (called the *EMA2* KO cross). Because proteins and mRNAs are exchanged between two mating pairs through the conjugation junction, *EMA2* mRNA/Ema2 protein expressed in the wild-type partner of the WT cross is expected to move into the *EMA2* KO partner and restore the *EMA2* KO phenotypes.

Total proteins were harvested at 4.5 and 6 hpm, and SUMOylated proteins were detected by western blotting using an anti-HA antibody (*Figure 4C*). At both time points, high molecular weight proteins (mainly >200 kDa) were detected in the WT cross (*Figure 4C*, WT), and they were reduced to ~50% in the *EMA2* KO cross (*Figure 4C*, KO). These results indicate that Ema2 is the major SUMO E3 ligase during the mid-conjugation stages. The remaining Ema2-independent SUMOylation is likely mediated by other SUMO E3 ligases (including the SP-RING containing proteins TTHERM_00227730, TTHERM_00442270 and TTHERM_00348490), and/or E3-independent SUMOylation (*Sampson et al., 2001*). The requirement of protein SUMOylation in DNA elimination was previously demonstrated in *Paramecium* by RNAi knockdown of *UBA2*, the gene encoding the SUMO E1 enzyme, and *SUMO* (*Matsuda and Forney, 2006*). Therefore, the involvement of a SUMO pathway in DNA elimination is likely conserved among ciliates.

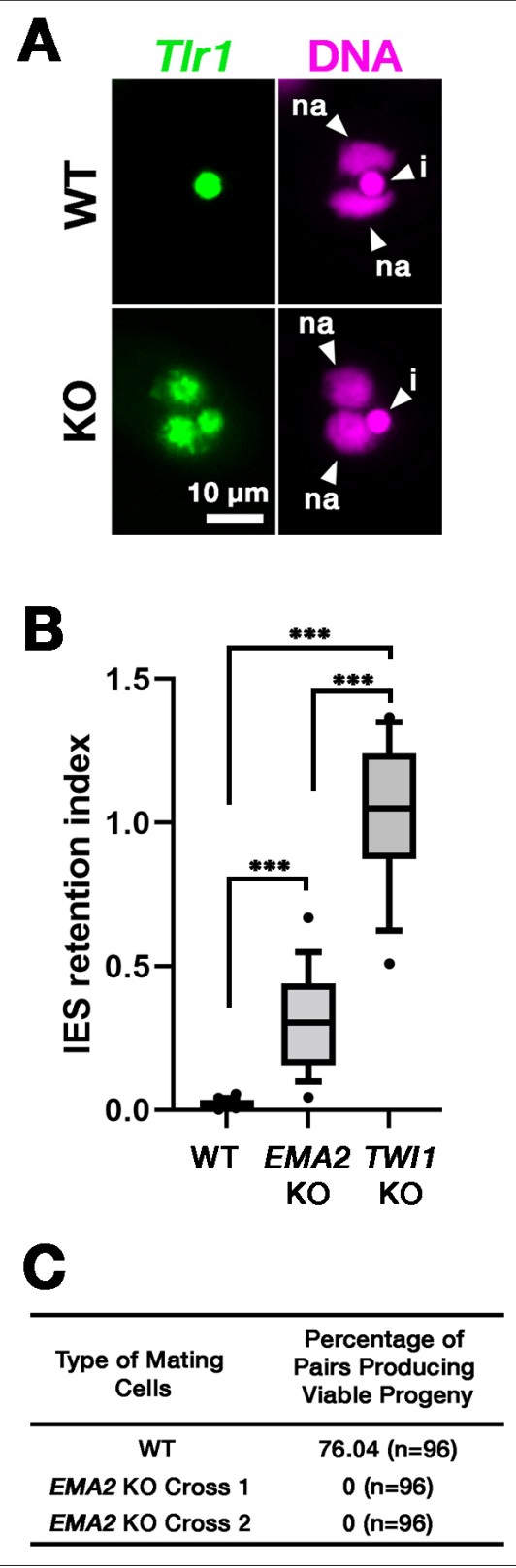

**Figure 2.** Ema2 is required for completing DNA elimination. (**A**) Two wild-type (WT) or two *EMA2* somatic KO (KO) cell lines were mixed, and their exconjugants at 36 hpm were analyzed by DNA-fluorescent in situ hybridization (FISH) with fluorescent probes complementary to the Tlr1 element (green). DNA was counterstained with DAPI (magenta). The micronucleus (MIC) and the new macronucleus (MAC) are marked with arrowheads with 'i' and

*Figure 2 continued on next page*

*Figure 2 continued*

'na', respectively. All pictures share the scale bar. (**B**) Exconjugants from wild-type (WT) cells, *EMA2* somatic KO cells, and *TWI1* KO cells were stained as in (**A**), the IES retention index was calculated (see Materials and Methods for details) from 20 cells each, and shown as box plots. The whiskers represent 10-90 percentile. Three asterisks (***) indicate a p-value of less than 0.001 in the Welch two-sample t-test. (**C**) Two wild-type (WT) cells and two independent crosses of *EMA2* somatic KO cells (Cross 1 and 2) were mated, and the conjugating pairs were isolated for the viability test. The percentages of pairs that gave rise to viable sexual progeny are shown. 'n' represents the number of total pairs tested.

## Ema2 is required for SUMOylation of Spt6

Next, to identify the SUMOylation target(s) of Ema2, we introduced a construct expressing His-tagged Smt3, which can also replace the essential function of Smt3 (*Figure 4—figure supplement 1* ), into an *EMA2* KO strain and crossed it with a wild-type strain (WT cross) or another *EMA2* KO strain (KO cross). Then, SUMOylated proteins at 6 hpm were concentrated using nickel-NTA beads and identified by mass spectrometry. We additionally examined the mating of wild-type strains without His-Smt3 expression and excluded any proteins identified with a log2 label-free quantification (LFQ) score above 25 or those possessing more than six consecutive histidine residues in this analysis, considering them as proteins binding to the nickel-NTA beads without His-Smt3 conjugation. Although most of the proteins were detected similarly between the WT cross and *EMA2* KO cross, Spt6, the most abundantly detected protein in the WT cross, was detected at a much lower level in the *EMA2* KO cross (*Figure 5A*). This result suggests that Spt6 is the major Ema2 target for SUMOylation.

To confirm the Ema2-dependent SUMOylation of Spt6, we introduced a construct expressing HA-tagged Spt6 (Spt6-HA) from the endogenous *SPT6* locus into an *EMA2* KO strain and mated it with a wild-type strain (WT cross) or another *EMA2* KO strain (*EMA2* KO cross). Then, total proteins harvested at 4.5 and 6 hpm were analyzed by western blotting using an anti-HA antibody (*Figure 5B*). In the WT cross, a slower migrating population of Spt6-HA was detected in addition to a band corresponding to unmodified Spt6-HA at both time points (*Figure 5B*, WT). In contrast, slower migrating Spt6-HA was barely detectable in the *EMA2* KO cross (*Figure 5B*, KO). Then, to examine the timing of the appearance of the slower migrating Spt6 species, we introduced the same Spt6-HA-expressing construct into a wild-type strain and Spt6-HA was analyzed by western blotting (*Figure 5—figure supplement 1*). Consistent with the Ema2-dependent appearance of the slower migrating Spt6-HA, they were not detected in growing and starved vegetative wild-type cells (*Figure 5—figure supplement 1*, Veg and 0 hpm, respectively) when Ema2 was not expressed (*Figure 1*). The slower migrating Spt6-HA was also detected at 8 hpm when the new MAC was already formed (*Figure 5—figure supplement 1*, 8 hpm) suggesting that Spt6 is possibly SUMOylated also in the new MAC.

The nature of slower migrating species of Spt6-HA was further examined by immunoprecipitating Spt6-HA using an anti-HA antibody. Among the total purified Spt6-HA (*Figure 5C*, left, IP-WT), the slower migrating species were detected by an anti-Smt3 antibody in the WT cross (*Figure 5C*, right, IP-WT), and such SUMOylated Spt6-HA species were greatly reduced in the *EMA2* KO cross (*Figure 5C*, KO). These results indicate that the slower migrating species of Spt6 are SUMOylated Spt6 and Ema2 is required for the majority of SUMOylation of Spt6 during the mid-conjugation stages. The remaining SUMOylation observed on Spt6 in the absence of Ema2 is likely facilitated by other SUMO E3 ligases and/or E3-independent SUMOylation, as discussed earlier for the other instances of Ema2-independent SUMOylations.

## Ema2 is required for the accumulation of lncRNA in the parental MAC

Spt6 is a conserved regulator of several steps of transcription in various eukaryotes. Because TDSD was proposed to be triggered by the base-pairing interaction between scnRNAs and nascent pMAC-lncRNAs in the parental MAC (*Aronica et al., 2008*; *Noto and Mochizuki, 2018*), we hypothesized that Ema2-dependent Spt6 SUMOylation promotes pMAC-lncRNA transcription.

To examine pMAC-lncRNAs, we amplified transcripts spanning IES-MDS borders by RT–PCR in which MAC-lncRNAs can be distinguished from MIC-lncRNAs by their lengths (*Figure 6A*). For the three loci examined, pMAC-lncRNAs were detected in wild-type cells but not in *EMA2* KO cells at 6 hpm, although the control *RPL21* mRNA was detected in both conditions (*Figure 6A*). The lack of detection of pMAC-lncRNAs in *EMA2* KO cells was not due to loss of the primer binding sites by

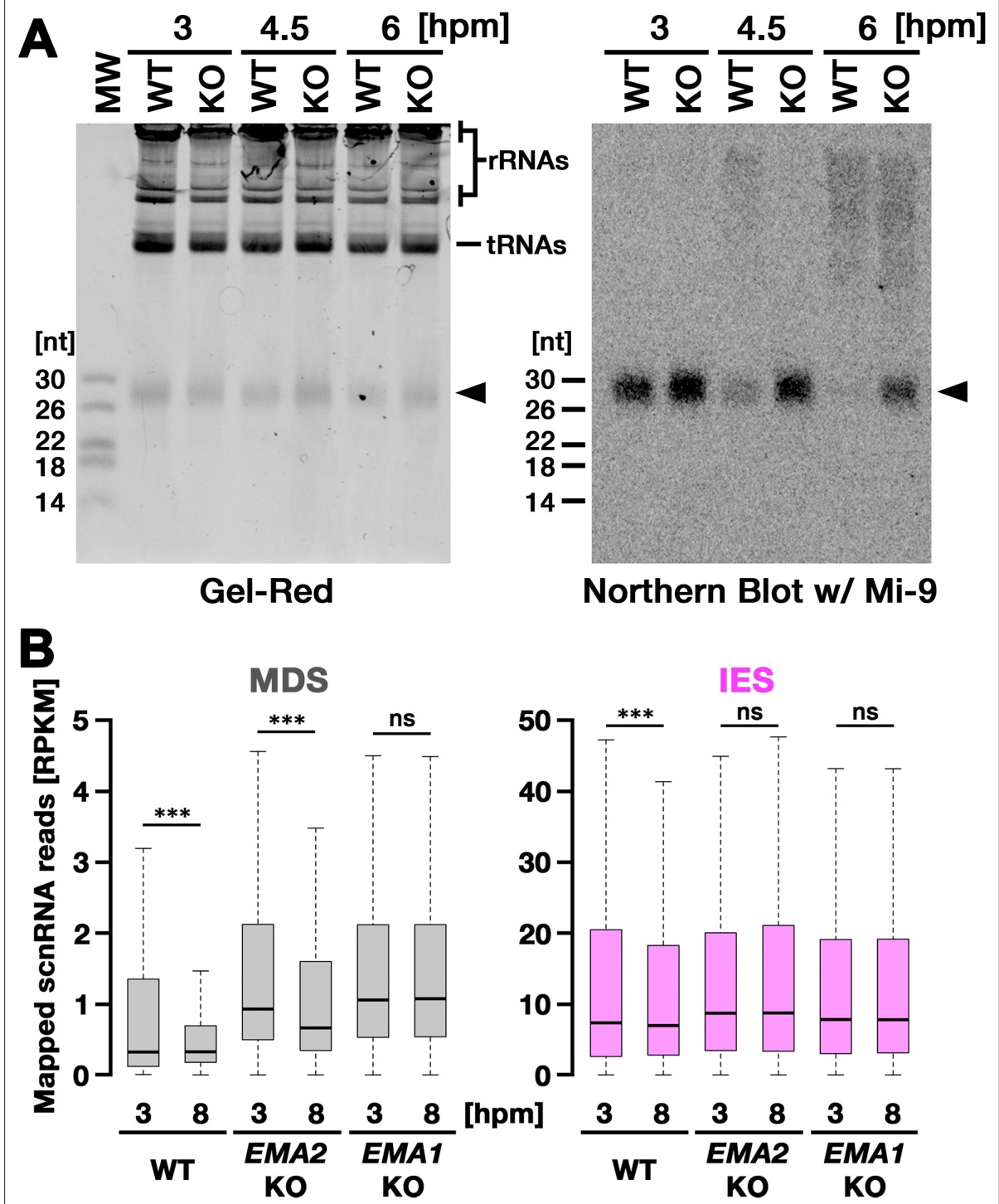

**Figure 3.** Ema2 is required for target-directed small RNAs (scnRNA) degradation (TDSD). (**A**) Total RNA was isolated from conjugating wild-type (WT) and *EMA2* somatic KO (KO) cells at 3, 4.5, and 6 hr post-mixing (hpm), separated in denaturing gel and stained with the nucleic acid dye Gel-Red (right). Then, RNA was transferred to a membrane and hybridized with the radioactive Mi-9 probe, which is complementary to a repetitive sequence in MDSs (right). (**B**) Small RNAs from conjugating wild-type (WT), *EMA2* somatic KO, and *EMA1* somatic KO strains were isolated at 3 and 8 hpm and analyzed by high-throughput sequencing. Normalized numbers (RPKM [read per kilobase of unique sequence per million]) of sequenced small RNAs (26–32 nt) that uniquely matched to the macronuclear-destined sequences (MDS) (left) or internal elimination sequence (IES) (right) genomic tiles (see Materials and methods) are shown as box plots. The median value is represented by the horizontal bar in the box. The minimum and maximum values are indicated by the bars on top and bottom of the box, respectively, with 1.5 x the interquartile range (IQR). Three asterisks (***) and 'ns' respectively indicate a p-value of less than 0.001 and more than 0.05 in the Wilcoxon rank sum test.

*Figure 3 continued on next page*

*Figure 3 continued*

The online version of this article includes the following source data for figure 3:

**Source data 1.** The raw data of northern blot (top) and Gel-red stained gel (bottom) without (Figure_3 A_Original) and with (Figure_3 A_Original-marked) marks of the positions of regions used for *Figure 3A*.

alternative DNA elimination in their prior sexual reproductions, as genomic PCR with the same primer sets detected the corresponding MAC loci in the *EMA2* KO cells (*Figure 6—figure supplement 1*). We, therefore, conclude that Ema2 is required for the accumulation of pMAC-lncRNAs at least for the three tested loci.

The accumulation of lncRNAs was also examined cytologically. As the consequence of the bidirectional lncRNA transcription in the three different nuclei, double-stranded RNAs are accumulated in

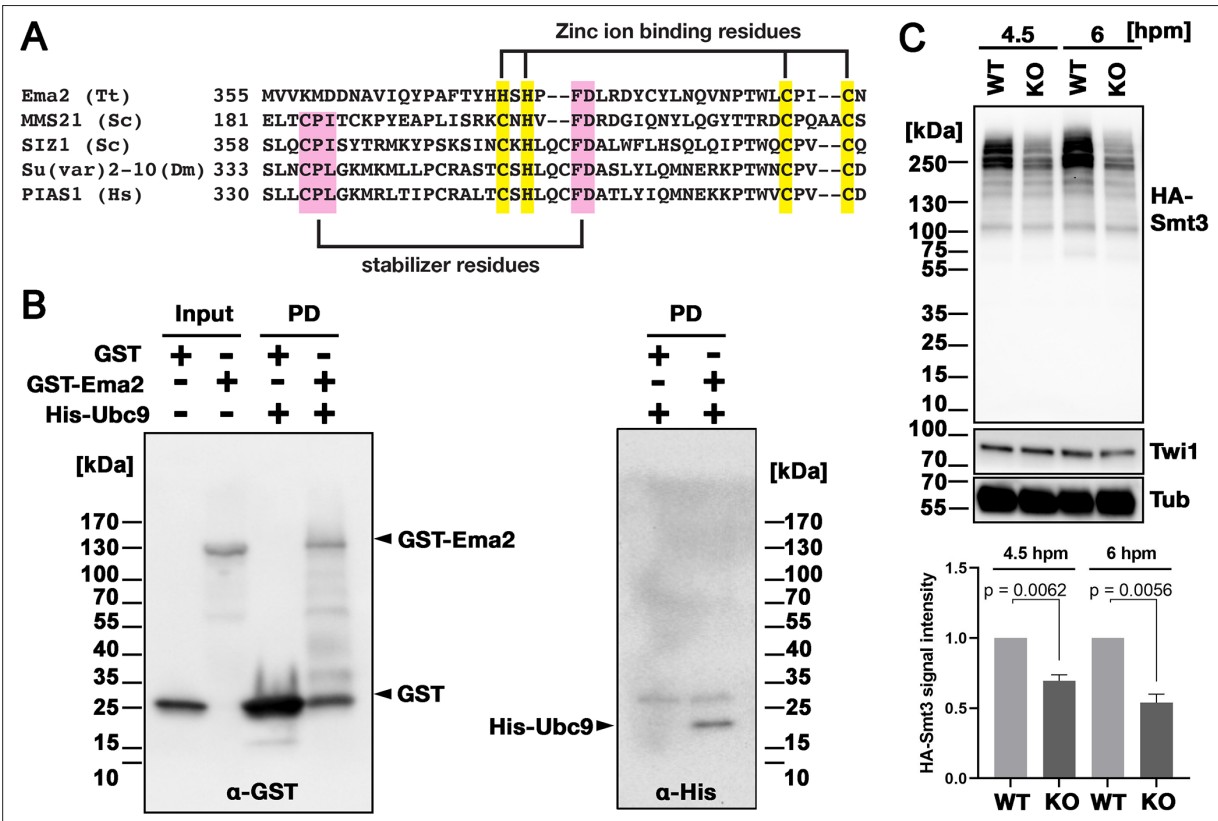

**Figure 4.** Ema2 acts as a SUMO E3 ligase. (**A**) The SP-RING domain of Ema2 is compared with that of MMS21 and SIZ1 in *S. cerevisiae*, Su(var)2–10 in *D. melanogaster* and PIAS1 in *H. sapiens*. The conserved cysteine and histidine residues that are involved in zinc ion binding are highlighted in yellow. The residues that stabilize the domain structure of some SP-RING domain proteins are marked with pink. (**B**) GST alone (GST), GST-tagged Ema2 (GST-Ema2), and His-tagged Ubc9 (His-Ubc9) were recombinantly expressed in *E. coli* and purified. GST and GST-Ema2 were immobilized on glutathione beads and incubated with His-Ubc9. Proteins retained on the beads were eluted, and the input and eluted proteins (PD) were analyzed by western blotting using anti-GST (left) and anti-His (right) antibodies. (**C**) An *EMA2* somatic KO strain expressing HA-tagged Smt3 was crossed with a wild-type strain (WT cross, WT) or another *EMA2* somatic KO strain (*EMA2* KO cross, KO), and their total proteins at the indicated time points were analyzed by western blotting using anti-HA (top), anti-Twi1 (middle) and anti-alpha tubulin (bottom) antibodies. The signal intensities of the anti-HA blots in the individual entire lanes were quantified in three independent experiments. The values in the WT cross were normalized to 1, and their means and standard deviations are presented as a bar graph, with p-values determined by the Welch two-sample t-test.

The online version of this article includes the following source data and figure supplement(s) for figure 4:

**Source data 1.** The raw data of western blot without (Figure_4B_Original) and with (Figure_4B_Original-marked) marks of the positions of regions used for *Figure 4B*.

**Source data 2.** The raw data of western blot without (Figure_4 C_Original) and with (Figure_4 C_Original-marked) marks of the positions of regions used for *Figure 4C*.

**Figure supplement 1.** HA-tagged Smt3 (HA-Smt3) and His-Smt3 can replace the essential function of Smt3.

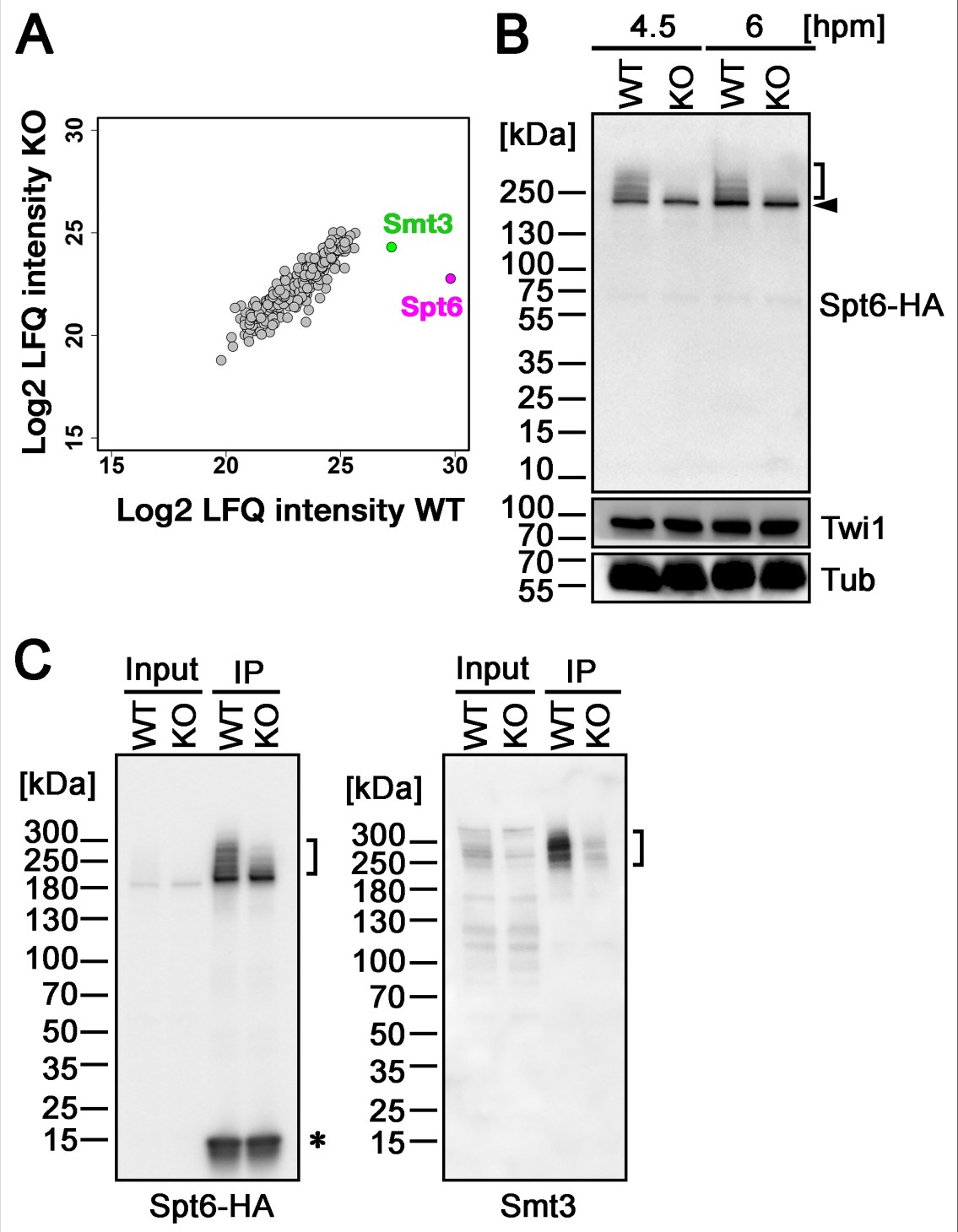

**Figure 5.** Ema2 promotes SUMOylation of Spt6. (**A**) A construct expressing His-tagged Smt3 (His-Smt3) was introduced into a wild-type and an *EMA2* somatic KO strain and crossed with another wild-type (WT cross) or *EMA2* somatic KO strain (*EMA2* KO cross). Proteins were harvested at 6 hpm, and His-Smt3-conjugated proteins were purified with Ni-NTA beads from conjugating cells and analyzed by mass spectrometry. Values of log base 2 of label-free quantification (LFQ) intensities of each identified protein between the WT cross (WT) and *EMA2* KO cross (KO) were compared. (**B**) A construct expressing HA-tagged Spt6 (Spt6-HA) was introduced into an *EMA2* somatic KO strain and crossed with a wild-type (WT-cross, WT) or another *EMA2* somatic KO (KO-cross, KO) strain. Total proteins were harvested at 4.5 and 6 hpm, and Spt6-HA was detected by western blotting using an anti-HA antibody. Twi1 and alpha-tubulin (Tub) were also analyzed to monitor mating efficiency and loading, respectively. The positions of modified and unmodified Spt6-HA proteins are marked with a bracket and an arrowhead, respectively. (**C**) Total proteins were harvested from WT and KO crosses at

*Figure 5 continued on next page*

*Figure 5 continued*

6 hpm (input), and Spt6-HA was immunoprecipitated using an anti-HA antibody (IP). The purified proteins were analyzed by western blotting using an anti-HA (left) or an anti-Smt3 (right) antibody. SUMOylated Spt6 proteins are marked with brackets. Unidentified protein cross-reacting with the anti-HA antibody is marked with an asterisk.

The online version of this article includes the following source data and figure supplement(s) for figure 5:

**Source data 1.** The raw data of western blot without (Figure_5B_Original) and with (Figure_5B_Original-marked) marks of the positions of regions used for *Figure 5B*.

**Source data 2.** The raw data of western blot without (Figure_5 C_Original) and with (Figure_5 C_Original-marked) marks of the positions of regions used for *Figure 5C*.

**Figure supplement 1.** Slower migrating Spt6 species appear during conjugation.

**Figure supplement 1—source data 1.** The raw data of western blot without (Figure_5-figure supplement_1_Original) and with (Figure_5-figure supplement_1_Original-marked) marks of the positions of regions used for *Figure 5—figure supplement 1*.

these nuclei (*Aronica et al., 2008*; *Malone et al., 2005*; *Mochizuki and Gorovsky, 2005*; *Woo et al., 2016*). As previously reported, immunostaining of wild-type cells using the long double-stranded (ds) RNA-specific J2 antibody (*Schönborn et al., 1991*; *Figure 6B*, WT) detected long dsRNAs first in the MIC at its premeiotic stage (3 hpm), then in the parental MAC at the mid stages of conjugation (6 hpm), and finally in the new MAC at late conjugation (9 hpm). In the *EMA2* KO cells, long dsRNAs were detected in the MIC at 3 hpm and in the new MAC at 9 hpm but were undetectable in the parental MAC at 6 hpm (*Figure 6B*, *EMA2* KO), indicating that Ema2 is specifically required for the accumulation of pMAC-lncRNAs.

In contrast, *EMA1* was dispensable for the accumulation of dsRNAs in the parental MAC (*Figure 6B*, *EMA1* KO), which is consistent with the previous notions that Ema1 is required for pMAC-lncRNAs to interact with scnRNAs but not their transcription (*Aronica et al., 2008*). Although it is unclear whether lncRNAs are single or double-stranded when Ema1 promotes the lncRNA-scnRNAs interaction, the less severe TDSD defect observed in the *EMA2* KO cells compared to the *EMA1* KO cells (*Figure 3B*) indicates that certain Ema1-dependent TDSD may be initiated by single-stranded lncRNAs or mRNAs that are transcribed independently of Ema2.

In parallel to TDSD, scnRNA-dependent accumulation of H3K27me3, which is catalyzed by the histone methyltransferase Ezl1 within PRC2, occurs in the parental MAC (*Liu et al., 2007*). We found that H3K27me3 was undetectable in the parental MAC in the *EMA2* KO cells at 6 hpm or in the *EMA1*, *TWI1,* and *EZL1* KO cells (*Figure 6C*). Therefore, Ema2 is required for the scnRNA-directed deposition of H3K27me3 in the parental MAC, which can be explained by the loss of the chromatin recruitment of PRC2 by the scnRNA-lncRNA interaction. Altogether, these results indicate that Ema2 is required for the genome-wide accumulation of pMAC-lncRNAs.

## Ema2 facilitates the chromatin association of Spt6 and RNA polymerase II

Next, we analyzed the localization of Spt6 by immunofluorescence staining. In both WT and *EMA2* KO cross described above, Spt6-HA was similarly detected in the MAC at 4.5 hpm (*Figure 7A*, left) and in the new MAC at 9 hpm (*Figure 7—figure supplement 1*). We also found that the MAC localization of Rpb3, the third largest subunit of RNA polymerase II (RNAPII) was not affected in the absence of Ema2 (*Figure 7A*, right). Therefore, at least at the cytological level, the absence of Ema2 does not affect the localization of Spt6 and RNAPII.

We then examined their subnuclear localization by cell fractionation (*Figure 7B*, left), in which cells at 4.5 hpm were first incubated with lysis buffer to extract soluble cyto- and nucleoplasmic proteins, and then, the insoluble fraction was sonicated to extract chromatin-bound proteins (*Ali et al., 2018*). In the WT cross (*Figure 7B*, WT), Spt6-HA was detected in both the soluble (S1) and chromatin-bound (S2) fractions, but slower-migrating SUMOylated Spt6-HA was detected only in the latter fraction, indicating that SUMOylated Spt6 is associated with chromatin. A similar distribution was detected for Rpb3. This chromatin association of Spt6 and RNAPII does not require RNA, as they were retained in the insoluble fraction after RNase A treatment without sonication (*Figure 7C*), although Twi1, which interacts with chromatin through nascent pMAC-lncRNA (*Aronica et al., 2008*), was mostly detected in the soluble fraction after the treatment (*Figure 7C*). In the *EMA2* KO cross, Spt6-HA and Rpb3 in

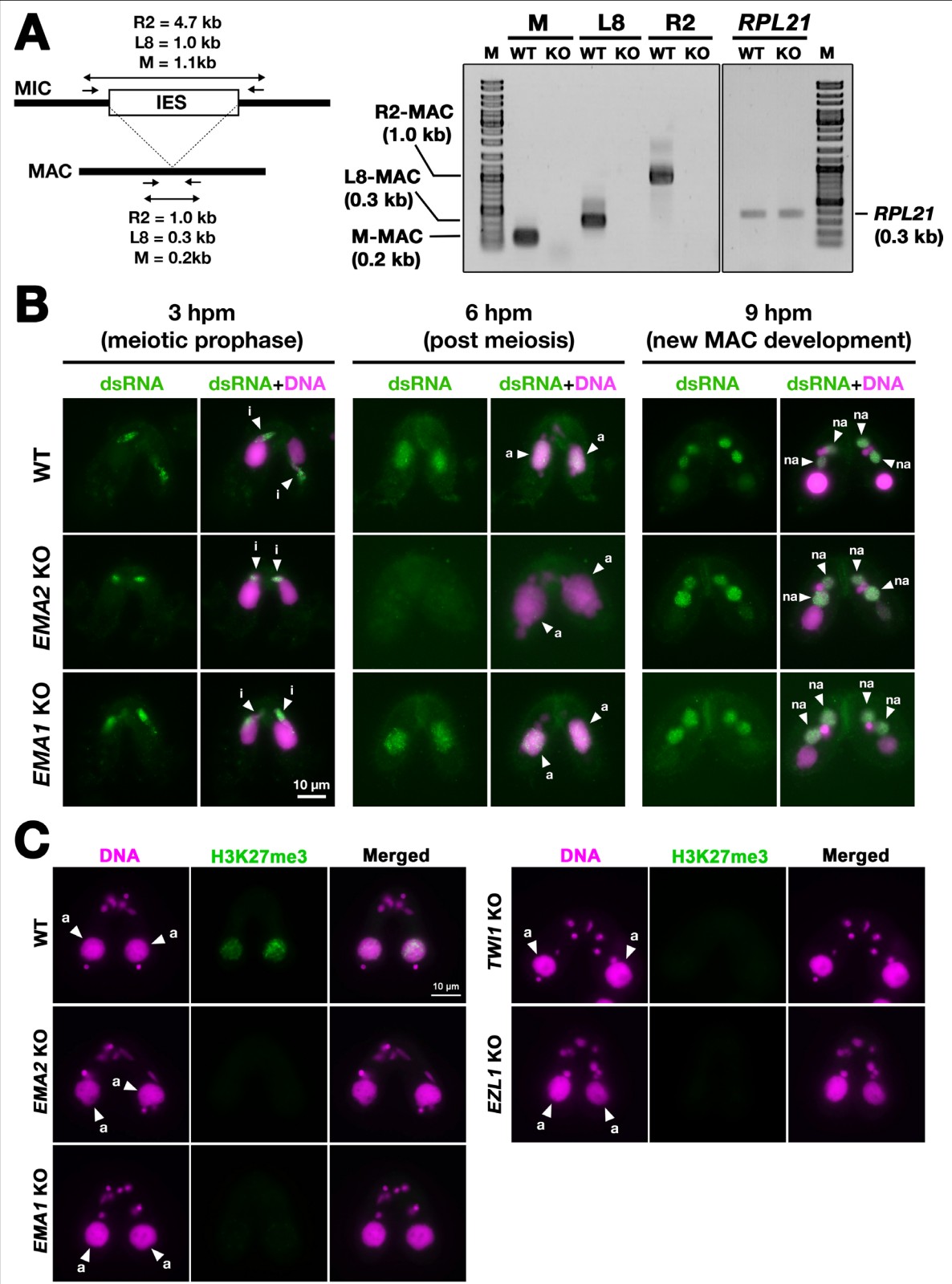

**Figure 6.** Ema2 is required for the accumulation of long non-coding RNA (lncRNA) transcripts from the parental macronucleus (MAC). (**A**) (Left) Schematic representation of the RT–PCR assay. The black bars and the open box represent the macronuclear-destined sequence (MDS) and internal elimination sequence (IES), respectively. The arrows represent the primers used for RT–PCR. The lengths of the PCR amplicons are shown with double-sided arrows. (Right) Wild-type (WT) or *EMA2* somatic KO (KO) cells were mated, and their total RNAs at 6 hpm were used for RT–PCR. The positions

*Figure 6 continued*

corresponding to the PCR products of lncRNAs from the MAC M, L8, and R2 loci are marked. *RPL21* mRNA was also analyzed as a positive control. (**B**) Conjugating wild-type (WT), EMA2 somatic KO, and EMA1 somatic KO cells at the indicated time points were analyzed by immunofluorescence staining using the anti-long dsRNA antibody J2 (dsRNA, green). DNA was counterstained with DAPI (magenta). The MIC, the parental MAC, and the new MAC are indicated with arrowheads with 'i,', 'a,'and '"na,' respectively. All pictures share the scale bar. (**C**) Heterochromatin formation in the parental MAC. Conjugating wild-type (WT), EMA2 somatic KO, EMA1 somatic KO, TWI1 complete (somatic + germline) KO, and EZL1 somatic KO cells at 6 hpm were analyzed by immunofluorescence staining using an anti-H3K27me3 antibody (green). DNA was counterstained with DAPI (magenta). The micronucleus (MIC) and the parental MAC are marked with arrowheads with 'i' and 'a,' respectively. All pictures share the scale bar.

The online version of this article includes the following source data and figure supplement(s) for figure 6:

**Source data 1.** The raw data of ethidium bromide-stained PCR products separated in agarose gels without (Figure_6 A_Original) and with (Figure_6 A_Original-marked) marks of the positions of regions used for *Figure 6A*.

**Figure supplement 1.** Confirmation of the presence of the corresponding genomic loci that were used to detect long non-coding RNAs (lncRNAs).

**Figure supplement 1—source data 1.** The raw data of ethidium bromide-stained PCR products separated in agarose gels without (Figure_6-figure supplement 1-Original) and with (Figure_6-figure supplement 1-Original-marked) marks of the positions of regions used for *Figure 6—figure supplement 1*.

---

the chromatin-bound fraction were greatly reduced (*Figure 7B*, KO, S2). These findings were further validated in two additional replicated experiments (*Figure 7—figure supplements 2 and 3*). We, therefore, conclude that Ema2 promotes the association of Spt6 and RNAPII with chromatin in the parental MAC at the mid-conjugation stage.

## Spt6 is not necessary to be SUMOylated in Ema2-directed lncRNA transcription

Next, to directly examine the importance of SUMOylation of Spt6, we intended to produce a SUMOylation-defective Spt6 mutant. *Tetrahymena* Spt6 consists of an N-terminal domain of low complexity (DOL) and a downstream region that is conserved among eukaryotes (*Figure 8A*, WT). We produced constructs expressing HA-tagged Spt6 with lysine to arginine substitutions either for all lysine residues in the DOL (*HA-SPT6-DOL-KR*) or for lysine residues that were on the surface of an AlphaFold predicted *Tetrahymena* Spt6 structure in one of the three non-DOL parts (N, M, and C in *Figure 8A*) in addition to those that were also predicted to be SUMOylation targets by the JASSA algorithm (*Beauclair et al., 2015*) in all three parts (*HA-SPT6-N-KR, HA-SPT6-M-KR, and HA-SPT6-C-KR, Figure 8A*).

We introduced these mutant constructs as well as the construct expressing wild-type Spt6 (*HA-SPT6-WT*) into *SPT6* KO cells and found that all of them were able to restore the lethality of *SPT6* KO cells (*Figure 8—figure supplement 1*). However, the cells rescued by *HA-SPT6-N-KR* and *HA-SPT6-M-KR* showed severe defects in meiotic progression and mating initiation, respectively, making their SUMOylation status during conjugation uninvestigable. In contrast, the cells rescued by *HA-SPT6-DOL-KR* and *HA-SPT6-C-KR* showed a seemingly normal progression of conjugation, although the *HA-SPT6-C-KR* cells showed lower mating efficiency. We found that while HA-Spt6-DOL-KR was SUMOylated like HA-Spt6-WT, SUMOylation was not detected on HA-Spt6-C-KR (*Figure 8B*). Therefore, Spt6-C-KR represents a SUMOylation-defective Spt6 mutant, exhibiting at least a reduced level of SUMOylation compared to Spt6 in the absence of Ema2 (compare *Figure 8B* and *Figure 5B*).

Cell fractionation experiments showed that HA-Spt6 and RNAPII were detected in the chromatin-bound fraction in *HA-SPT6-C-KR* cells as in the *HA-SPT6-WT* cells consistently in three independent experiments (*Figure 8C*, *Figure 8—figure supplement 2*), and long dsRNAs were detected in the parental MAC (*Figure 8D*) as well as in the new MAC (*Figure 8—figure supplement 3*) in the *HA-SPT6-C-KR* cells. These results suggest that, contrary to our expectation, Spt6 SUMOylation per se is not required for the lncRNA transcription in the parental MAC. Nonetheless, exconjugants from the *HA-SPT6-C-KR* cells showed a mild DNA elimination defect (*Figure 8E*), indicating that Ema2-directed Spt6-SUMOylation plays a role in DNA elimination other than promoting pMAC-lncRNA transcription.

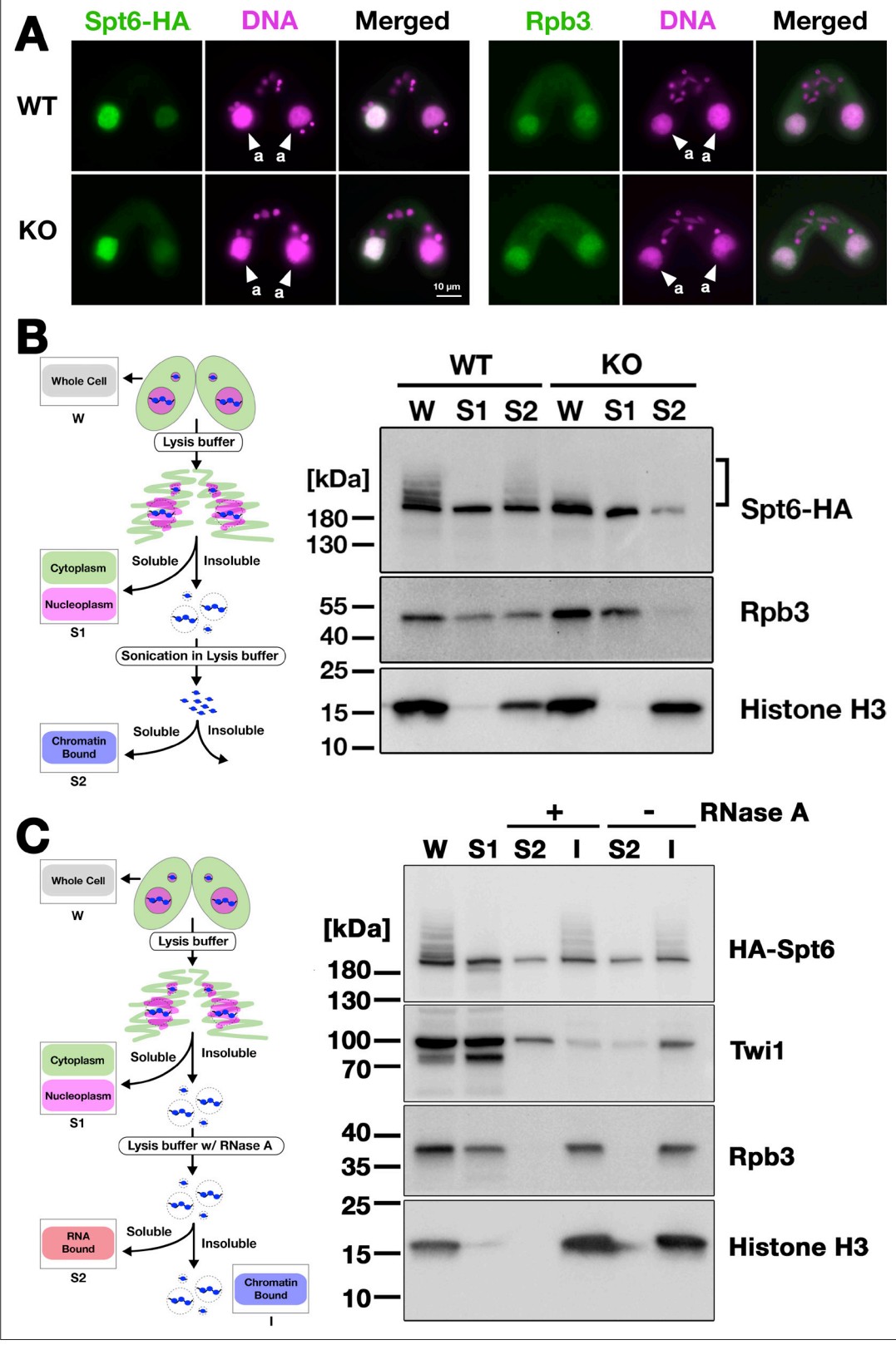

**Figure 7.** Ema2 facilitates the chromatin association of Spt6 and RNA polymerase II. (**A**) A construct expressing HA-tagged Spt6 (Spt6-HA) was introduced into an *EMA2* somatic KO strain and crossed with a wild-type (WT-cross, WT) or another *EMA2* somatic KO (KO-cross, KO) strain. The localizations of Spt6-HA and Rpb3, the third largest subunit of RNA polymerase II (RNAPII), were analyzed by immunofluorescence staining at 4.5 hpm using anti-HA

*Figure 7 continued on next page*

*Figure 7 continued*

(left) or anti-Rpb3 (right) antibodies, respectively. DNA was counterstained with DAPI (magenta). The parental macronucleus (MACs) are marked with arrowheads with 'a.' All other structures stained by DAPI are micronucleus (MICs). All pictures share the scale bar. (**B**) (Left) Schematic representation of the cell fractionation assay. Conjugating cells were incubated with a lysis buffer that releases cytoplasmic (green) and nucleoplasmic (magenta) proteins to the soluble fraction (**S1**). Then, the insoluble fraction was resuspended in fresh lysis buffer and sonicated. The solubilized fraction after sonication (**S2**) contains fragmented chromatin. (Right) S1 and S2 fractions as well as total cellular proteins (**W**) from WT-cross (WT) and *EMA2* KO-cross (KO) explained in (**A**) at 4.5 hpm were analyzed by western blotting using anti-HA (Spt6-HA), anti-Rpb3, and anti-histone H3 antibodies. The position of SUMOylated Spt6-HA is marked with a bracket. (**C**) (Left) Schematic representation of the cell fractionation assay with RNase treatment. Conjugating cells were incubated with a lysis buffer that releases cytoplasmic (green) and nucleoplasmic (magenta) proteins to the soluble fraction (**S1**). Then, the insoluble fraction was incubated in fresh lysis buffer with (+) or without (-) RNase A, and the solubilized (**S2**) and insoluble (**I**) fractions were obtained. (Right) S1, S2, and I fractions as well as total cellular proteins (**W**) from *SPT6* germline (MAC + MIC) KO strains rescued with the *HA-SPT6-WT* construct (see *Figure 8*) were analyzed by western blotting using anti-HA (HA-Spt6), anti-Twi1p, anti-Rpb3, and anti-histone H3 antibodies. The position of SUMOylated HA-Spt6 is marked with a bracket.

The online version of this article includes the following source data and figure supplement(s) for figure 7:

**Source data 1.** The raw data of western blot without (Figure_7B-Original) and with (Figure_7B-Original-marked) marks of the positions of regions used for Figure_7B.

**Source data 2.** The raw data of western blot without (Figure_7C-Original) and with (Figure_7C-Original-marked) marks of the positions of regions used for Figure_7 C.

**Figure supplement 1.** Localization of Spt6 at late conjugation stage.

**Figure supplement 2.** Replicated cell fractionation assay examining the chromatin association of proteins.

**Figure supplement 2—source data 1.** The raw data of western blot without (Figure_7-figure supplement 2-Original) and with (Figure_7-figure supplement 2-Original-marked) marks of the positions of regions used for *Figure 7—figure supplement 2*.

**Figure supplement 3.** Replicated cell fractionation assay examining proteins associated with the chromatin in an RNA-dependent manner.

**Figure supplement 3—source data 1.** The raw data of western blot without (Figure_7-figure supplement 3-Original) and with (Figure_7-figure supplement 3-Original-marked) marks of the positions of regions used for *Figure 7—figure supplement 3*.

## Discussion

In this study, we showed that the conjugation-specific SUMO E3 ligase Ema2 is required for the accumulation of lncRNAs (*Figure 6*), TDSD (*Figure 3A and B*), and heterochromatin formation (*Figure 3C*) in the parental MAC and eventually for completing DNA elimination (*Figure 1*) in *Tetrahymena*. We found that Ema2 is responsible for SUMOylation of the transcriptional regulator Spt6 (*Figure 5*) and promotes the interaction of Spt6 and RNAPII with chromatin (*Figure 7*). We, therefore, conclude that Ema2 facilitates genome-wide lncRNA transcription in the parental MAC, which is a prerequisite for scnRNA-chromatin communication and thus for the downstream TDSD that regulates DNA elimination.

Our observation that Ema2 enhances the chromatin interaction of Spt6 and RNAPII (*Figure 7*) seems to contradict the fact that Spt6 and RNAPII are essential for vegetative cell viability (*Figure 8— figure supplement 1*; *Mochizuki and Gorovsky, 2004a*), but Ema2 is expressed exclusively during conjugation (*Figure 2*). Moreover, as *EMA2* KO cells did not significantly impede the progression of conjugation processes, any essential mRNA transcriptions for these processes must take place in the parental MAC during conjugation even in the absence of Ema2. Therefore, the observed loss of the majority of Spt6 and RNAPII from chromatin in the absence of Ema2 (*Figure 7B*) must be a temporal event during the mid-conjugation stage. This suggests that Spt6 and RNAPII might be specifically engaged in pMAC-lncRNA transcription at this particular time window in wild-type cells. It is likely that some radical change in the chromatin environment in the parental MAC is required for the global transcription of pMAC-lncRNAs, and Ema2-dependent SUMOylation might maintain Spt6 and RNAPII on chromatin for transcription in such an environment. Alternatively, because the expression of many of the transcriptional machineries, including Spt6 and RNAPII subunits, is highly upregulated during conjugation (*Miao et al., 2009*; *Mochizuki and Gorovsky, 2004a*; *Xiong et al., 2012*), the

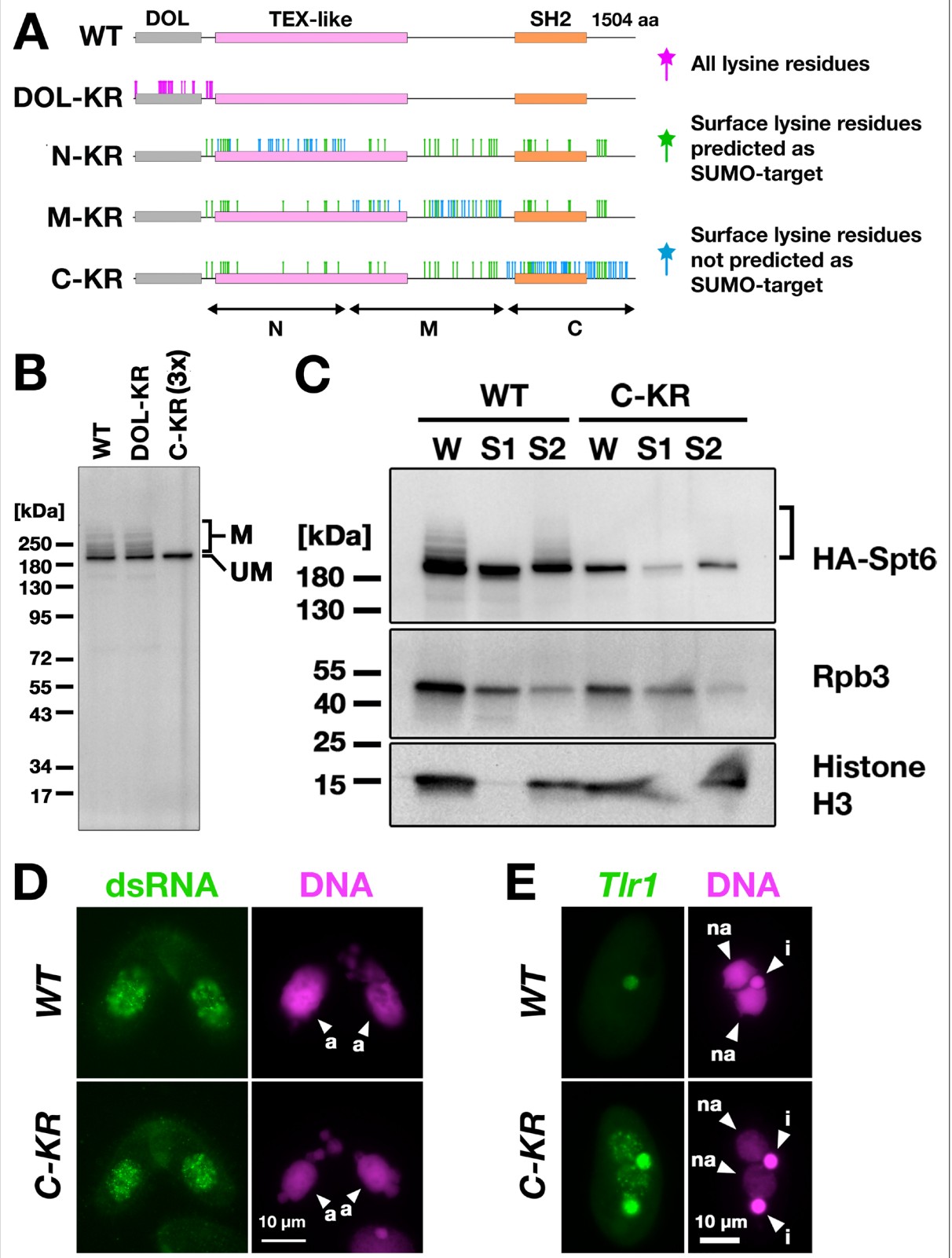

**Figure 8.** Spt6 SUMOylation is dispensable for parental macronucleus (MAC) long non-coding RNA (lncRNA) transcription. (**A**) Schematic representation of *SPT6* mutant constructs with lysine (K)-to-arginine (R) substitutions. Spt6 has a domain of low complexity (DOL) region followed by conserved TEX-like and SH2 domains. The DOL-KR mutant has K to R substitutions for all lysine residues in the DOL region. N-KR, M-KR, and C-KR mutants have K to R substitutions for lysine residues that were on the surface of a predicted Spt6 structure in one of the three non-DOL regions (N, M, and C-regions)

*Figure 8 continued on next page*

*Figure 8 continued*

in addition to those surface lysine residues that were also predicted to be SUMOylatable by an algorithm in all three regions. (**B**) Total proteins were harvested at 4.5 hpm from conjugating *SPT6* germline (MAC + MIC) KO cells rescued with wild-type (*HA-SPT6-WT*), *HA-SPT6-DOL-KR,* or *HA-SPT6-C-KR* constructs. Because mating was low in the *HA-SPT6-C-KR* rescued cells, three times (3 x) more total protein sample was loaded for these cells. HA-Spt6 was detected by western blotting using an anti-HA antibody. The positions of modified and unmodified HA-Spt6 proteins are marked with 'M' and 'UM,' respectively. (**C**) *HA-SPT6-WT* (*WT*) and *HA-SPT6-C-KR* (*C–KR*) rescued cells at 4.5 hpm were analyzed by cell fractionation as described in *Figure 7B*. The position of SUMOylated HA-Spt6 is marked with a bracket. (**D**) Accumulation of long dsRNAs in *HA-SPT6-WT* (*WT*) and *HA-SPT6-C-KR* (*C–KR*) rescued cells at 4.5 hpm was analyzed by immunofluorescence staining using the J2 antibody as described in *Figure 6B*. The parental MACs are marked with arrowheads with 'a.' All other structures stained by DAPI are micronucleus (MICs). All pictures share the scale bar. (**E**) *HA-SPT6-WT* (*WT*) and *HA-SPT6-C-KR* (*C–KR*) rescued cells at 36 hpm were analyzed by DNA- FISH with fluorescent probes complementary to the Tlr1 element (red). DNA was counterstained with DAPI (magenta). The MIC and the new MAC are marked with arrowheads with 'i' and 'a,' respectively. All pictures share the scale bar.

The online version of this article includes the following source data and figure supplement(s) for figure 8:

**Source data 1.** The raw data of western blot without (Figure_8B-Original) and with (Figure_8B-Original-marked) marks of the positions of regions used for *Figure 8B*.

**Source data 2.** The raw data of western blot without (Figure_8C-Original) and with (Figure_8C-Original-marked) marks of the positions of regions used for *Figure 8C*.

**Figure supplement 1.** Genetic rescue of *SPT6* KO.

**Figure supplement 2.** Replicated cell fractionation assay examining the effect of the SUMOylation-defective Spt6 mutation.

**Figure supplement 2—source data 1.** The raw data of western blot without (Figure_8-figure supplement 2-Original) and with (Figure_8-figure supplement 2-Original-marked) marks of the positions of regions used for *Figure 8—figure supplement 2*.

**Figure supplement 3.** Localization of long non-coding RNA (lncRNA) transcripts in the SUMOylation-defective SPT6 mutant at late conjugation stage.

transcription of pMAC-lncRNAs might require a greater amount of transcriptional machinery than other types of transcription, and Ema2-dependent SUMOylation might be required for committing these excess machineries to chromatin for transcription.

Our investigation of the SUMOylation-defective Spt6 mutant suggested that although Spt6 is SUMOylated in an Ema2-dependent manner (*Figure 5*), SUMOylation of Spt6 by itself is not required for pMAC-lncRNA transcription (*Figure 8*). This observation suggests that Ema2 may facilitate pMAC-lncRNA transcription through SUMOylation of some protein(s) other than Spt6. Although, we failed to detect any other protein that was SUMOylated in an Ema2-dependent manner (*Figure 5A*), some Ema2-dependent SUMOylation events might not be detected in our current approach if Ema2-dependent and Ema2-independent SUMOylation occur in the same proteins and/or if Ema2-dependent SUMOylation results in protein degradation, such as through SUMO-targeted ubiquiti-nation (*Praefcke et al., 2012*; *Staudinger, 2017*). Alternatively, SUMOylation of Spt6 may promote pMAC-lncRNA transcription by competing with some other modification that occurs at the same lysine residue(s) and our SUMOylation-defective Spt6 mutant might actually mimic the SUMOylated state of Spt6 by preventing such modification. To further investigate the role of Ema2, future studies will need to identify individual Ema2-dependent SUMOylated lysine residues under conditions of in vivo proteasome inhibition and to compare the status of other modifications of Spt6 in the presence and absence of Ema2. These are also important to understand the role of Spt6 SUMOylation in DNA elimination (*Figure 8E*).

It was reported that SUMOylation of the heterochromatin components Swi6, Chp2, and Clr4 is important for RNAi-directed heterochromatin silencing in fission yeast (*Shin et al., 2005*). Addition-ally, SUMOylation of the histone deacetylase HDAC1 promotes small RNA-directed transcriptional silencing in *C. elegans* (*Kim et al., 2021*). Furthermore, in the piRNA-directed transposon silencing of *Drosophila*, the SUMO E3 ligase Su(var)2–10 plays an essential role in the recruitment of the histone methyltransferase complex SetDB1, which deposits H3K9me3 (*Ninova et al., 2020a*; *Ninova et al., 2020b*), and Su(var)2–10-independent SUMOylation of Panoramix (Panx) also promotes transcrip-tional silencing by recruiting the general heterochromatin effector Sov (*Andreev et al., 2022*). All these SUMOylation events seem to require pre-existing lncRNA transcription. In contrast, this study showed that Ema2-dependent SUMOylation during the process of DNA elimination in *Tetrahymena* occurs upstream of lncRNA transcription (*Figure 6A and B*). In addition, the fact that the loss of the Piwi protein Twi1 blocks the accumulation of H3K27me3 in the parental MAC (*Liu et al., 2007*; *Xu et al., 2021*) without affecting lncRNA accumulation (*Woo et al., 2016*) indicates that heterochromatin

formation is not a prerequisite for lncRNA transcription in *Tetrahymena*, in contrast to heterochromatin-dependent lncRNA transcription of piRNA clusters in *Drosophila* (*Andersen et al., 2017*). Altogether, *Tetrahymena* likely uses a unique evolutionary solution for transcribing lncRNAs to promote TDSD and small RNA-directed heterochromatin formation. We, therefore, believe that further investigations of the role of Ema2-dependent SUMOylation in lncRNA transcription will clarify an undescribed layer of regulatory mechanisms in small RNA-directed chromatin regulation.

## Materials and methods

### Strains and cell culture condition

The *Tetrahymena thermophila* wild-type strains B2086, CU427, and CU428 and the MIC defective 'star' strains B*VI and B*VII were obtained from the *Tetrahymena* stock center. The production of the *EMA2* KO strains was described previously (*Shehzada and Mochizuki, 2022*). The other transgenic *Tetrahymena* strains are described below. The *Tetrahymena* strains reported in this manuscripts are available from the corresponding author upon reasonable request. Cells were grown in SPP medium (*Gorovsky et al., 1975*) containing 2% proteose peptone at 30 °C. For conjugation, growing cells (~5–7 × 10$^5$ /mL) of two different mating types were washed, pre-starved (~12–24 hr), and mixed in 10 mM Tris (pH 7.5) at 30 °C.

### Oligo DNA, synthetic DNA, and antibodies

The primers and synthetic DNAs used in this study are listed in *Supplementary file 1*. The following antibodies were obtained from manufacturers: anti-H3K27me3 (Millipore, 07–449, RRID:AB_310624); anti-HA HA11 (Covance, clone 6B12, RRID:AB_291231); anti-H3K9me3 (Active Motif, 39162); anti-GST (BD Biosciences, 554805, RRID:AB_395536); anti-His (Proteintech, 66005–1-Ig, RRID:AB_11232599); anti-long dsRNA J2 (Jena Bioscience RNT-SCI-10010200, RRID:AB_2651015); anti-α-tubulin 12G10 (Developmental Studies Hybridoma Bank of the University of Iowa). The anti-Rpb3 antibody was described previously (*Kataoka and Mochizuki, 2017*). The anti-Smt3 antibody was a gift from Dr. James Forney (Purdue University) and described previously (*Nasir et al., 2015*).

### Establishment of *EMA2*-HA strains

The C-terminal part of the *EMA2* coding sequence was amplified from the genomic DNA of B2086 using EMA2-C-FW and EMA2-HA-C-RV. Similarly, the 3' flanking sequence of *EMA2* was amplified using EMA2-F-FW and EMA2-F-RV. The two PCR products were combined by overlapping PCR using EMA2-C-FW and EMA2-F-RV and cloned into XbaI and XhoI sites of pBlueScriptSK (+) to obtain pEMA2. Next, the *HA-neo3* construct was excised out from pHA-neo3 (*Kataoka et al., 2010*) using SalI and BamHI and cloned into the same restriction sites of pEMA2 to obtain pEMA2-HA-neo3. Finally, pEMA2-HA-neo3 were digested with XbaI and XhoI and introduced into the *EMA2* MAC locus of B2086 and CU428 by homologous recombination using biolistic transformation (*Bruns and Cassidy-Hanley, 2000*) followed by the selection with 100 µg/mL paromomycin and 1 µg/mL CdCl$_2$. Then, the endogenous *EMA2* MAC copies were replaced by the *EMA2-HA-neo3* locus through phenotypic assortment (*Hamilton and Orias, 2000*).

### Immunofluorescent staining and western blotting

Immunofluorescent staining of long dsRNA was performed as described (*Woo et al., 2016*). The other immunofluorescent staining experiments were performed as described (*Loidl and Scherthan, 2004*). For immunofluorescent staining, all the primary and respective secondary antibodies (Alexa-488-coupled anti-rabbit or mouse IgG, Invitrogen) were diluted 1:1000. For western blotting, all the primary and respective secondary antibodies (HRP-coupled anti-rabbit or mouse IgG, Jackson ImmunoResearch Lab) were diluted 1:10,000.

### DNA elimination assay by FISH

DNA elimination assay by FISH was performed as described (*Loidl and Scherthan, 2004*). The plasmid DNAs pMBR 4C1, pMBR 2, and Tlr IntB, which contain different parts of the Tlr1element were used to produce Cy5-labeled FISH probes by nick translation (*Wuitschick et al., 2002*). Images of stained cells were analyzed by Fiji software. Whole areas of the new MAC (A) and MIC (I) and a part of the

cytoplasm (C) were manually encircled in the images; and the intensities of DAPI and Cy5 (FISH) signals in each area were measured. Then, the IES retention index was calculated as follows: 0.35 × {[(mean Cy3 intensity A - mean Cy3 intensity C)]/[(mean Cy3 intensity I - mean Cy3 intensity C)]}/ {[(mean DAPI intensity A - mean DAPI intensity C) ]/[(mean DAPI intensity I - mean DAPI intensity C)]}. Based on our previous observation, DNA elimination is completely blocked in the absence of *TWI1* KO, therefore, the correction factor of 0.35 was obtained experimentally to make the average IES retention index of *TWI1* KO cells to 1.

## Viability test for progeny

To test the progeny viability, mating pairs were isolated into drops of 1 x SPP at 6 hpm and incubated for ~3 days. Then, the completion of the conjugation of grown-up cells from wild-type and *EMA2* KO pairs were examined for their 6-methylpurine (6-mp) resistance and paromomycin sensitivity, respectively as described previously (*Mochizuki et al., 2002*).

## Small RNA analyses

For northern hybridization analysis of scnRNAs, 10 µg of total RNAs isolated from conjugating wild-type and *EMA2* KO cells using TRIzol reagent were separated in a 15% acrylamide-urea gel and analyzed by northern blot using the radio-labeled Mi-9 probe as previously described (*Aronica et al., 2008*). The hybridized probe was detected by Typhoon IP Phosphorimager (GE healthcare). High-throughput sequencing and analyses of small RNAs were performed as described (*Mutazono et al., 2019*). The 2020 version of the MAC genome assembly (*Sheng et al., 2020*) was fragmented into 10 kb pieces. Each 10 kb fragment that contains longer than 3 kb mappable sequence was used as an MDS tile (total 10,235 tiles). Each Type-A IES (*Noto et al., 2015*) was used as an IES tile (total of 4691 tiles). Normalized numbers (RPKM [reads per kilobase of unique sequences per million]) of 26- to 32-nt small RNAs that uniquely matched one of the MDS and IES tiles were counted.

## GST pull down assay

*E. coli* codon-optimized synthetic *EMA2* and *UBC9* genes (*EMA2-Ec and UBC9-Ec*) were cloned into the BamHI and XhoI sites of pGEX-4T-1 and NdeI and BamHI sites of pET28a (+), respectively to obtain pGEX-4T-1-EMA2-Ec and pET28-UBC9-Ec. pET28-UBC9-Ec was introduced into BL21 *E. coli* strain, His-Ubc9 expression was induced ~16 hr at 25 °C with 0.05 mM isopropylthio-β-galactoside (IPTG), the cells were lysed in HEPES buffer (50 mM HEPES pH 7.5, 800 mM KCL, 10% glycerol, 0.2 mM PMSF, 1 x cOmplete ETDA-free [Roche]) including 10 mM imidazole, incubated with Ni-NTA beads for 2 hr at 4 °C, washed four times with HEPES buffer including 20 mM imidazole, eluted with HEPES buffer including 250 mM imidazole, dialyzed with 50 mM Tris pH 7.5, 150 mM NaCl, 2 mM MgCl$_2$, 1 mM DTT, 50% glycerol and finally passed through Microcon 10 kDa filter (Merck). For GST pull-down assay, BL21 *E. coli* cells were transformed with pGEX-4T-1 or pGEX-4T-1-*EMA2-Ec,* and expression of GST-alone or GST-Ema2 was induced ~16 hr at 25 °C, lysed with 50 mM Tris–HCl, pH 7.5, 100 mM NaCl, 1 mM DTT, 0.2 mM PMSF, 1x cOmplete EDTA-free and incubated with Glutathione Sepharose 4B for 90 min at 4 °C. Then, the beads were incubated with 10 µg of His-Ubc9 in GST-pull down buffer (20 mM Tris–HCl, pH 7.5, 100 mM NaCl, 1% NP-40, 5 mM MgCl$_2$, 1 mM DTT, 0.2 mM PMSF, 1 x cOmplete EDTA-free) for 90 min at 4 °C, washed four times with GST-pull down buffer and eluted with 2 x SDS sample buffer. The eluted proteins were analyzed by western blotting.

## Functionality test of *HA-SMT3* and *His-SMT3* using *SMT3* germline KO strains

The *SMT3* homozygous heterokaryon KO strain GC6 (*Nasir et al., 2015*) was obtained from Dr. James Forney (Purdue University). This strain was crossed with B2086, and sexual progeny were selected for their paromomycin resistance. After sexual maturation, clones showing paromomycin sensitivity (i.e. the *SMT3* KO locus was removed from the MAC by phenotypic assortment) were established as *SMT3* heterozygous heterokaryon KO strains. Then they were crossed with B*VII to obtain their round I exconjugants. Finally, their genotypes were examined by PCR to select two *SMT3* homozygous heterokaryon KO strains, B1-1 and B5-17. They were then mated and pBCMB1-His-SMT3 (or pBCMB1-His for negative control) or pBCS3B1-HA-SMT3 (or pBCS3B1-HA for negative control) were introduced into the new MAC at 8 hpm by particle gun. The cells were incubated in 10 mM Tris pH 7.5 at 30 °C

until 24 hpm and fed by adding the equal volume of 2x SPP. After 3 hr incubation at 30 °C, the *neo* expression (and His-Smt3 expression for pBCMB1-His-SMT3) was induced by adding 1 µg/mL CdCl2 and cells were incubated an additional 1 hr. Then 100 µg/mL paromomycin was added and aliquoted to 96-well plates (150 µL/well, seven plates). The plates were incubated at 30 °C for 4–5 days and paromomysin-resistant cells were further examined by genomic PCR to confirm genetic rescue by the introduced constructs (but not by the endogenous *SMT3* copy from the parental MAC).

## Establishment of HA-Smt3 and His-Smt3 expressing strains

*STM3* cDNA was amplified by RT-PCR from the total RNA of vegetatively growing CU428 using SMT3_HA_FW and SMT3_HisExt_RV and cloned into the BamHI and SpeI sites of pBNMB1-HA using NEBuilder HiFi DNA assembly kit to obtain pBNMB1-HA-SMT3. Next, neo5 was excised with SalI and XmaI and replaced with pur6, which was amplified from pBP6MB1 with BTU1-LCFW and MTT1_5UTR-SeqRV, using NEBuilder HiFi DNA assembly kit to obtain pBP6MB1-HA-SMT3. For His-SMT3 expression, the HA-tag encoding sequence of pBP6MB1-HA-SMT3 was replaced by a 6x His tag-encoding sequence. Then, pBP6MB1-HA-SMT3 and pBP6MB1-His-SMT3 were digested by XhoI and introduced into the MAC *BTU1* locus of the *EMA2* KO cells. Transformation, selection, and phenotypic assortment were performed as described for the *SPT6-HA* strains.

## Affinity purification of SUMOylated proteins

For purification of His-Smt3-conjugated proteins, *EMA2* KO cells containing the BP6MB1-His-SMT3 construct were mated with either a wild-type or another *EMA2* KO strain and His-SMT3 was induced by adding 0.05 µg/mL CdCl$_2$. The purification was performed as descried (*Hendriks and Vertegaal, 2016*; *Liebelt et al., 2019*). Briefly, the cells were lysed in Guanidine Lysis Buffer (6 M Guanidine-HCl, 93.2 mM Na$_2$HPO$_4$, 6.8 mM NaH$_2$PO$_4$, 10 mM Tris-HCl pH 8.0) by sonication (2 × 25 times, 20% power, 60% pulse) using a probe sonicator (Omni-Ruptor 250 Ultrasonic Homogenizer). After centrifugation, the cleared lysate was supplemented with 50 mM imidazole and 5 mM β-mercaptoethanol and incubated with Ni-NTA agarose beads. The beads were washed, and the bound proteins were eluted with Elution buffer (6 M Urea, 58 mM Na$_2$HPO$_4$, 42 mM NaH$_2$PO$_4$, 10 mM Tris-HCl pH 8.0, 500 mM imidazole). The eluted proteins were precipitated in 10% TCA, resuspended in 1 x SDS buffer, and analyzed by western blot and mass-spectrometry. Wild-type cells that do not express His-Smt3 was also analyzed in parallel to identify proteins that intrinsically bind to the Ni-NTA agarose beads. For purification of HA-Smt3-conjugated proteins, total proteins were precipitated by incubating cells in 6% trichloroacetic (TCA) on ice for 10 min and then spinned down at 13,000 rpm for 5 min at 4 °C. The precipitate was dissolved in 1 x SDS sample buffer, neutralized with 2 M Tris, and then incubated for 5 min at 95 °C. The dissolved proteins (600 µl) were then suspended in 7.4 mL of 20 mM Tris pH 7.5, 100 mM NaCl, 2 mM MgCl$_2$, 2 mM CaCl$_2$ and incubated with Ezview Red anti-HA agarose beads (Sigma) for overnight at 4 °C. Then the beads were washed five times with 20 mM Tris pH 7.5, 100 mM NaCl, 2 mM MgCl$_2$, 2 mM CaCl$_2$, 0.1% tween 20 and the bound proteins were eluted by incubating with 1 x SDS buffer for 5 min at 95 °C. The eluted proteins were analyzed by western blotting.

## Establishment of Spt6-HA expressing strains

To obtain the *SPT6*-HA construct, a part of the coding sequence of *SPT6* was amplified from the genomic DNA of B2086 using *SPT6*-FW1-SacII and *SPT6*-RV2-overlap. Similarly, the 3' flanking sequence was amplified using *SPT6*-3F-FW2-overlap and *SPT6*-3F-RV1-KpnI. The two PCR products were combined by overlapping PCR using *SPT6*-FW1-SacII and *SPT6*-3F-RV1-KpnI and cloned into the SacII and KpnI sites of pBlueScriptSK (+) to obtain p*SPT6*. Next, the NheI and XhoI fragment of pHA-neo3 was cloned into the same restriction sites of p*SPT6* to obtain p*SPT6-HA-neo3*. Then, *neo3* was excised from p*SPT6*-HA-neo3 with XmaI and SalI and replaced by *pur6* from pBP6MB1-TR-TUBE to obtain p*SPT6-HA-pur6*. Finally, the p*SPT6-HA-pur6* plasmid were digested by SacII and KpnI and introduced into the MAC *SPT6* locus of an *EMA2* KO and a wild-type strain by biolistic transformation. The transformed cells were selected in 300 µg/mL puromycin (Invitrogen) and 1 µg/mL CdCl$_2$ and cells were phenotypically assorted as previously described (*Hamilton and Orias, 2000*), until cells grew in 1.2 mg/mL of puromycin.

## RT-PCR analyses of lncRNA transcripts

Total RNA from mating wild-type and *EMA2* KO cells were isolated at 6 hpm using TRIzol reagent (Invitrogen). The RNA was treated with Turbo DNase (Ambion) and cDNA was synthesized from 1 μg of RNA using SuperScript II (Invitrogen) with random 6-mer as primers. The lncRNA transcripts from the MAC loci containing the M-, L8-, and R2 IES boundaries (see also *Figure 6A*) were amplified by nested PCR (95 °C for 20 s/50 °C for 30 s/68 °C for 1 min; 30 cycles each) using the primers M5'–3+M3'–3 (M-1st), M5'–4+M5'–4 (M-2nd), L8_5'–1+L8_3'–1 (L8-1st), L8_5'–2+L8_3'–2 (L8-2nd), R2_5'–1+R2_3'–1 (R2-1st), and R2_5'–2+R2_3'–2 (R2-2nd) and analyzed in 1% agarose gel. As a positive control, the constitutively expressed *RPL21* mRNA was detected with RPL21-FW and RPL21-RV.

## Cell fractionation

Cell fractionation was performed as described (*Ali et al., 2018*). 1.4 x 10⁶ cells were incubated with 1 mL of Lysis buffer (50 mM Tris pH 8.0, 100 mM NaCl, 5 mM MgCl2, 1 mM EDTA, 0.5% Triton X-100, 1 x Complete Protease Inhibitor) on ice for 30 min and spun down at 15,000 g for 10 min at 4 °C. The proteins in 500 μL of supernatant were precipitated by adding 10% (final) TCA, washed three times with ice-cold acetone, and dissolved in 70 μL 1x SDS sample buffer, while the precipitate was dissolved in 2 mL of fresh Lysis buffer, sonicated (five times of 20% power-50% pulse-5 pulses with >20 s interval) using a probe sonicator (Omni-Ruptor 250 Ultrasonic Homogenizer), and then centrifuged at 20,000 g for 10 min at 4 °C. Proteins in 1 mL of the supernatant (solubilized chromatin) were precipitated by adding 10% (final) TCA, washed three times with ice-cold acetone and dissolved in 70 μL 1x SDS sample buffer. For cell Fractionation with RNase treatment, $1.4 \times 10^6$ cells were incubated with 1 mL of Lysis buffer containing 20 μg/mL RNase A on ice for 30 min and spun own at 15,000 g for 10 min at 4 °C. The proteins in 1 mL of the supernatant were precipitated by adding 10% (final) TCA, washed three times with ice-cold acetone, and dissolved in 70 μL 1x SDS sample buffer, while the precipitate was directly dissolved in 140 μL of 1 x SDS sample buffer.

## Establishment of *SPT6* germline KO strains

To create the knockout construct for *SPT6*, the 5' and 3' flanking regions of the *SPT6* gene were amplified by PCR using the primer sets SPT6-KO-5F-FW/SPT6-KO-5F-RV and SPT6-KO-3F-FW/SPT6-KO-3F-RV, respectively and concatenated by overlap PCR using SPT6-KO-5F-FW and SPT6-KO-3F-RV. The PCR product was cloned into the XhoI and BamHI digested pBlueScriptSK(+). Then the obtained plasmid was digested with BamHI and the pm-resistant cassette *neo4*, which was isolated from the pNeo4 plasmid (*Aronica et al., 2008*) by SmaI digestion (*Bruns and Cassidy-Hanley, 2000*), was inserted. All assembly reactions were done using NEBuilder HiFi DNA assembly mix (NEB). The resulting plasmid, pSPT6-KO-neo4, was digested with SacII and XhoI to release the targeting construct from the vector backbone and introduced into the MIC of mating UMPS214 and UMPS811 cells (*Vogt and Mochizuki, 2013*) by biolistic transformation. Heterozygous progenies were selected with 100 μg/mL pm and 500 μg/mL 5-Fluoroorotic Acid (5-FOA, Fermentas) in the presence of 1 μg/ml CdCl₂. The resulting heterozygous strains were cultured for 10 passages without pm and 5-FOA, and pm-sensitive (pm-s) heterozygous strains were isolated. The pm-s heterozygous cells were then mated with B*VI or B*VII to obtain their round I exconjugants. Finally, their genotypes were examined by PCR to select *SPT6* homozygous heterokaryon KO strains.

## Establishment of wild-type and mutant *SPT6* strains

*SPT6* cDNA was amplified from total RNA of mating B2086 and CU428 at 6 hpm with SPT6-FW-Turbo and SPT6-RV-Turbo, cloned into SpeI and BamHI sites of pBCMB1-HA-Turbo to obtain pBCMB1-HA-Turbo-SPT6. Then, the *MTT1* promoter and the HA-Turbo-tag were removed using AgeI and XmaI and replaced by the *SPT6* promoter and the N-terminal HA tag, using the overlapping product of two genomic PCR fragments amplified with SPT6-5F-FW, SPT6-5F-RV and with SPT6-N-FW-HA and SPT6-N-RV to obtain pBCSB-HA-SPT6. To produce pBCSB-HA-SPT6-DOL-KR, the SPT6 cDNA lacking the first 248 amino acids were amplified using SPT6_M249_FW and SPT6_M249_RV, replaced with the NdeI-BamHI fragment of pBCSB-HA-SPT6 and then the synthetic DNA SPT6-DOL-KtoR was inserted into the BamHI site. To make pBCSB-HA-SPT6-N-KR, pBCSB-HA-SPT6-M-KR and pBCSB-HA-SPT6-C-KR, the conserved region of *SPT6* was excised from pBCSB-HA-SPT6 using NdeI and AgeI and replaced respectively with Synthetic DNA Set 1 (SPT6-gBlock-118KR-1, SPT6-gBlock-42KR-2,

and SPT6-gBlock-42KR-3), Synthetic DNA Set 2 (SPT6-gBlock-42KR-1, SPT6-gBlock-118KR-2, and SPT6-gBlock-42KR-3), or Synthetic DNA Set 3 (SPT6-gBlock-42KR-1, SPT6-gBlock-42KR-2, and SPT6-gBlock-118KR-3). All assembly reactions were done using NEBuilder HiFi DNA assembly mix (NEB). The constructs were digested with XhoI and introduced into the *BTU1* locus of the new MAC of conjugating cells of two *SPT6* KO homozygous heterokaryon strains at 10 hpm by electroporation. Progeny rescued by the introduced constructs were selected in 100 µg/mL paromomycin and 1 µg/mL CdCl$_2$, and cells were phenotypically assorted until cells grew in ~100–120 µg/mL cycloheximide.

## Acknowledgements

We acknowledge the MRI facility, member of the national infrastructure France-BioImaging supported by the French National Research Agency (ANR-10-INBS-04), the Proteomics Platform (PPM) of BioCampus Montpellier, and the NGS unit of Vienna BioCenter Core Facilities. This work was supported by an Advanced Grant from the 'Investissements d'avenir' program Labex EpiGenMed (ANR-10-LABX-12–01) and an 'Accueil de Chercheurs de Haut Niveau' grant (ANR-16-ACHN-0017) from the French National Research Agency, Equipes a FRM 2022 grant from the Fondation Recherche pour Médicale (FRM, EQU202203014651), an ARC 2021 PJA3 grant from the ARC Foundation (ARCPJA2021060003830) to KM, and a 'Fin de these' program fellowship from Fondation pour la Recherche Médicale (FRM, FDT20210601285) to SS.

## Additional information

### Funding

| Funder | Grant reference number | Author |
|---|---|---|
| Agence Nationale de la Recherche | ANR-10-INBS-04 | Kazufumi Mochizuki |
| Agence Nationale de la Recherche | ANR-10-LABX-12-01 | Kazufumi Mochizuki |
| Agence Nationale de la Recherche | ANR-16-ACHN-0017 | Kazufumi Mochizuki |
| Fondation pour la Recherche Médicale | FDT20210601285 | Salman Shehzada |
| Fondation pour la Recherche Médicale | EQU202203014651 | Kazufumi Mochizuki |
| Fondation ARC | ARCPJA2021060003830 | Kazufumi Mochizuki |

The funders had no role in study design, data collection and interpretation, or the decision to submit the work for publication.

### Author contributions

Salman Shehzada, Data curation, Formal analysis, Validation, Investigation, Visualization, Methodology, Writing – original draft, Writing – review and editing; Tomoko Noto, Investigation, Methodology, Writing – original draft, Writing – review and editing; Julie Saksouk, Investigation, Project administration; Kazufumi Mochizuki, Conceptualization, Resources, Data curation, Formal analysis, Supervision, Funding acquisition, Validation, Investigation, Visualization, Methodology, Writing – original draft, Project administration, Writing – review and editing

### Author ORCIDs

Salman Shehzada 
Tomoko Noto 
Julie Saksouk 
Kazufumi Mochizuki 

### Decision letter and Author response

Decision letter https://doi.org/10.7554/eLife.95337.sa1

Author response https://doi.org/10.7554/eLife.95337.sa2

## Additional files

### Supplementary files
- MDAR checklist
- Supplementary file 1. List of DNA oligos and synthethic DNAs.

### Data availability
The small RNA sequencing data have been deposited in the Gene Expression Omnibus (GEO) database (https://www.ncbi.nlm.nih.gov/geo) with the accession no. GSE243435. The mass spectrometry proteomics data have been deposited to the ProteomeXchange Consortium via the MassIVE partner repository (https://massive.ucsd.edu/) with the database identifier MSV000092977.

The following datasets were generated:

| Author(s) | Year | Dataset title | Dataset URL | Database and Identifier |
|---|---|---|---|---|
| Shehzada S, Noto T, Mochizuki K | 2023 | A SUMO E3 ligase promotes long non-coding RNA transcription to regulate small RNA-directed DNA elimination | http://www.ncbi.nlm.nih.gov/geo/query/acc.cgi?acc=GSE243435 | NCBI Gene Expression Omnibus, GSE243435 |
| Mochizuki K | 2023 | Comparison of SUMOylated proteins between wild-type and EMA2 KO Tetrahymena cells | https://massive.ucsd.edu/ProteoSAFe/dataset.jsp?task=3bf460eba84a44888e150146be16c688 | MassIVE partner repository, MSV000092977 |

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
