## [Editor Report]

This important study demonstrates that protein SUMOylation is essential for programmed DNA elimination guided by small RNAs during conjugation in *Tetrahymena* ciliates. The authors present convincing evidence that the E3 SUMO ligase Ema2 is necessary for the production of long non-coding RNAs from the somatic nucleus, targeted small RNA degradation, and DNA elimination. The authors also show that the transcription regulator Spt6 is a SUMOylation target of Ema2, though the relevance of this is not completely established. This paper is of broad significance and will appeal to those interested in non-coding RNA biology, the control of programmed genome rearrangements, or ciliate biology.

---

## [Decision Letter]

[Editors' note: this paper was reviewed by Review Commons.]

---

## [Author Response]

General Statements [optional]We would like to thank all reviewers for their valuable and constructive comments, which helped us a lot to improve the manuscript.

Point-by-point description of the revisions

Reviewer #1 :Reviewer #1 (Evidence, reproducibility and clarity (Required)):The authors convincingly show that Ema2, a conjugation-specific SUMO E3 ligase, localizes in the parental MAC during early conjugation stages, then moves to the new MAC. Using somatic EMA2 KO strains, they show that Ema2 is necessary for IES elimination and the recovery of viable progeny. They demonstrate that MAC scnRNAs do not disappear in an EMA2 KO and conclude that Ema2 is required for TDSD. They also show that ds lncRNA amounts in the parental MAC drop to background levels in an EMA2 KO, while they remain similar to WT in meiotic MICs or the new MACs.They also present evidence supporting that the transcription factor Spt6 is one of the targets of Ema2-mediated sumoylation. Spt6 is found in the parental MAC of conjugating cells, regardless of Ema2. However, Ema2 is crucial for the stable chromatin association of both Spt6 and Rpb3 (a subunit of RNA polymerase II). Unexpectedly, a non-sumoylatable Spt6 mutant is able to complement a SPT6 KO, since it maintains the synthesis of lncRNA in the parental MAC. Nonetheless, this mutant strongly impairs new MAC development and IES elimination. As a whole, the role of Spt6 sumoylation in programmed DNA elimination is not clearly established, and it probably affects another step than pMAC-lncRNA synthesis.Strong points

The demonstration that pMAC-lncRNA accumulation depends upon Ema2 is convincing. This finding provides novel insights into the mechanism involved in TDSD in Tetrahymena. An important point that would be worth discussing is how ds pMAC-lncRNAs may pair with scnRNAs.An RNA helicase (Ema1?) may play an important role in this process.

The requirement of Ema1 in the interaction between pMAC-lncRNAs and scnRNAs was reported previously by us (Aronica et al. 2008), which has been cited in this manuscript. Related to this point, we have added the following discussion in the revised manuscript (Page 10, Line 30): “Although it is unclear whether lncRNAs are single or double stranded when Ema1 promotes the lncRNA-scnRNAs interaction, the less severe TDSD defect observed in the *EMA2* KO cells compared to the *EMA1* KO cells (Figure 3B) indicates that certain Ema1-dependent TDSD may be initiated by single-stranded lncRNAs or mRNAs that are transcribed independently of Ema2”.

The manuscript is very well written. I noticed only a few typos (see minor comments below).

The pointed typos have been corrected in the revised manuscript.

The experiments are overall well done and well described. For non-Tetrahymena readers, it would be useful to clarify in the Results section (or in figure captions) whether the different KOs are in the MAC and/or also in the MIC

We have indicated whether each KO line is somatic or germline (MAC+MIC) in the figure legends whenever these lines are referenced.

The search for Ema2 targets using mass spectrometry was performed in a wild-type SMT3 background. This implies that endogenous wild-type Smt3 may have competed with His-Smt3 for protein sumoylation. To what extent may this have been a problem for the enrichment of sumoylated proteins on nickel columns? This point is critical, since the authors discuss that other proteins involved in pMAC-lncRNA transcription may be modified by Ema2 (p. 12). They should repeat the experiment in an SMT3 KO, or use anti-Smt3 antibodies to enrich for sumoylated proteins. If this is not possible, they should at least provide additional explanations.

We agree that a competition between His-tagged and non-tagged Smt3 lowered the sensitivity for the identification of SUMOylated proteins and we might miss some Ema2-dependent SUMOylated protein in the current study. However, we believe such protein, if any, is SUMOylated at very low level and not highly likely to be involved in the genome-wide orchestration of lncRNA transcription. We rather think that a critical Ema2-dependent SUMOylation event might be missed because some other residues of the same protein are SUMOylated by Ema2-independent manner and it was detected as a protein that was SUMOylated in both wild-type and *EMA2* KO condition. Therefore, as was explained in Discussion, it is important to identify individual residues that are SUMOylated in Ema2-dependent manner. We are on our way to set up an experimental system that allows us to detect individual SUMOylated residues in *Tetrahymena* and we hope to analyze the functions of Ema2-dependent SUMOylated residues in future studies.

In Figure 7A, the authors only show the localization of Spt6 in early exconjugants. Since Spt6 is essential for vegetative growth, one can expect that it also localizes in the vegetative MAC. Is it also found in the new developing MACs? The authors should complete the figure with additional panels showing vegetative cells and exconjugants at later stages (with their new MAC).

The Spt6 is indeed localized in the MAC during vegetative growth and in the new MAC at late conjugation stage in the wild-type condition. We did not detect any anomaly of Spt6 localization in the *EMA2* KO cells at least at the cytological level. The immunostaining results at the late conjugation stage are shown in Figure 7—figure supplement 1 in the revised manuscript and mentioned in the revised text (Page 11, Line 13). The immunostaining results of vegetatively growing cells are only attached below because Spt6 localization at vegetative stage when *EMA2* is not expressed is not highly relevant to this study.

Along the same line, the authors show that the non sumoylatable Spt6 mutant does not inhibit pMAC-lncRNA synthesis. No scnRNA analysis is shown under these conditions: does TDSD still take place? It would also be interesting to check whether lncRNAs are still produced in the new MACs.

The nonSUMOylatable Spt6 mutant (we now call SUMOylation defective Spt6 mutant according to one of the Reviewer 3’s suggestions) show lower mating, making us difficult to investigate its effect on TDSD. Because we did not detect Spt6 SUMOylation prior to mating, we believe the low mating phenotype of this mutant is not directly due to the loss of SUMOylation but instead some of the 77 K to R mutations affect the functions of Spt6 in efficient initiation of mating. Therefore, to precisely measure the effect of Ema2-dependent Spt6 SUMOylation, we need to identity exact Ema2-dependent SUMOylated residues of Spt6 to produce another nonSUMOylatable Spt6 mutant with fewer number of mutations that does not affect the mating process. Engaging in such work demands a substantial time investment, and we believe that the reviewers will concur that these experiments are components of our future projects.

Long dsRNA accumulation in the new MACs detected by the J2 antibody was comparable between wild-type and the SUMOylation-defective Spt6 mutant, suggesting that Spt6 SUMOylation is not necessary to produce lncRNAs in the new MAC. The data have been shown in Figure EV9 and mentioned in the main text (Page 12, Line 24) in the revised manuscript.

The experiment shown in Figure 4C indicates that high-molecular weight (possibly sumoylated) proteins decrease to 50% in the EMA2 KO: this suggests that another sumoylation activity exists in the cell. A search for other putative SUMO E3 ligases is missing in this study.

A few other putative SUMO E3 ligases indeed encoded in the *Tetrahymena* genome. Moreover, it is known that some substrates are SUMOylated without any SUMO E3 ligase in other eukaryotes. These points have been described in the revised text as follows (Page 8, Line 22): “The remaining Ema2-independent SUMOylation is likely mediated by other SUMO E3 ligases (including the SP-RING containing proteins TTHERM_00227730, TTHERM_00442270 and TTHERM_00348490) and/or E3-independent SUMOylation (Sampson et al. 2001).”

We agree that exploring the roles of other SUMO E3 ligases in *Tetrahymena* would be important and interesting, and we believe it will be one of our future projects.

Can one exclude that Spt6 is sumoylated at other stages (vegetative or during new MAC development) in an Ema2-independent manner?

We have now included western blot observation of Spt6 at different life stages of wild-type cells as Figure EV2. We did not detect any slower-migrating Spt6 species in vegetative cells. This has been mentioned in the revised text as follows (Page 9, Line 17):

“Then, to examine the timing of the appearance of the slower migrating Spt6 species, we introduced the same Spt6-HA-expressing construct into a wild-type strain and Spt6-HA was analyzed by western blotting (Figure EV2). Consistent with the Ema2-dependent appearance of the slower migrating Spt6-HA, they were not detected in growing and starved vegetative wild-type cells (Figure EV2, Veg and 0 hpm, respectively) when Ema2 was not expressed (Figure 1). The slower migrating Spt6-HA was also detected at 8 hpm when the new MAC was already formed (Figure EV2, 8 hpm) suggesting that Spt6 is possibly SUMOylated also in the new MAC.”

In which nucleus does coding transcription take place between 4.5 and 6 hpm? Can we exclude that the weaker association of Rpb3 with chromatin in the EMA2 KO cross also impairs coding transcription?

Coding transcription takes place in the parental MAC at 4.5 and 6 hpm in wild-type cells. Also, because *EMA2* KO cells did not show obvious defect in the progression of the conjugation processes, any essential mRNA transcriptions for these processes must occur even in the absence of Ema2. These points prompted us to add the following discussion in the Discussion section (Page 13, Line 14):

“Moreover, as *EMA2* KO cells did not significantly impede the progression of conjugation processes, any essential mRNA transcriptions for these processes must take place in the parental MAC during conjugation even in the absence of Ema2. Therefore, the observed loss of the majority of Spt6 and RNAPII from chromatin in the absence of Ema2 (Figure 7B) must be a temporal event during the mid-conjugation stage. This suggest that RNAPII might be specifically engaged in pMAC-lncRNA transcription at this particular time window in wild-type cells.”

The authors do not explain how they found Ema2. More information could be useful.

Ema2 was identified as a protein involved in DNA elimination during our systematic genetic investigation of genes exclusively expressed during conjugation. This has been mentioned in the revised manuscript (Page 6, Lines 4-5).

In Figures 2B and 3B: the statistical significance of the differences observed for the IES retention index and small RNA amounts should be evaluated using appropriate tests.

The result shown in Figure 2B (IES retention analysis) has been tested by Welch two-sample ttest and outcomes have been shown in the revised Figure 2B.

The result shown in Figure 3B (small RNA seq) has been tested by Wilcoxon rank sum test and outcomes have been shown in the revised Figure 3B.

Figure 3 caption: define acronym "IQR".

The definition of IQR (the interquartile range) has now been mentioned in the figure legend in the revised manuscript.

Figure 5 caption (line 4): there may be a word missing ("from conjugating cells?")

We have corrected the sentence by adding “cells” after “from conjugating” in Page30-Line 34.

Figure 8C: what does the asterisk stand for?

We realized that the asterisk is not necessary in the figure and thus it have been removed in the revised figure.

p. 10 (bottom): an "o" is missing in "Aronica et al. 2008".

We have corrected the error.

p. 13 (2nd line): remove final "s" in "mimic".

We have corrected the error.

p. 14: change "were" to "was" in "the production of the EMA2 KO strains was described previously"

We have corrected the error.

p. 14: remove capital letters in "Gorovsky"

We have corrected the error.

p. 15 (Viability test for progeny): what does "6-mp" stand for?

It is 6-methylpurine. We have added this information to the revised manuscript.

p. 17 (end of first paragraph): change "contracts" to "constructs".

We have corrected the error.

p. 17 (2nd line of last paragraph): change "was" to "were " in "EMA2 cells containing the BP6MB1His-SMT3 construct were mated…"

We have corrected the error.

p. 19 (3rd line of 2nd paragraph"): "spined own" should be replaced by "spinned down".

We have corrected the error.

Reviewer #1 (Significance (Required)):In this manuscript Shehzada et al. report important novel findings on the molecular mechanisms involved in RNA-mediated control of programmed DNA elimination in the ciliate *Tetrahymena thermophila*. In this organism, non-coding transcription takes place in distinct nuclei and produces double-stranded (ds) long non-coding RNAs (lncRNAs) at different stages during conjugation. First, bidirectional transcription in the MIC during meiosis produces ds lncRNAs that are processed to short scnRNAs. Second, lncRNAs from the parental MAC (pMAC-lncRNAs) are thought to drive the degradation of scnRNAs homologous to parental MAC DNA, in a process called TDSD (target-directed scnRNA degradation). Third, the remaining MIC-specific scnRNAs are imported to the new MACs, where their pair with lncRNAs and drive heterochromatin formation and DNA elimination.The present study focuses on TDSD, a process that has been poorly described at the molecular level. The strongest part of the work is the demonstration that the SUMO E3 ligase Ema2 is necessary for the production of pMAC-lncRNAs, which in turn impairs the selective degradation of MAC scnRNAs. A less convincing part is the identification of Ema2 targets. The authors identify Spt6 as one of the Ema2-dependent sumoylated proteins. However, they show that Spt6 sumoylation is not necessary for pMAC-lncRNA transcription.In principle, the results presented in this manuscript should be of broad interest for the scientific communities working on non-coding RNA biology and the epigenetic control of programmed genome rearrangements.Reviewer #2Reviewer #2 (Evidence, reproducibility and clarity (Required)):SummaryDuring conjugation (the sexual reproduction stage in the Tetrahymena ciliates), programmed DNA elimination guided by small RNAs termed scnRNAs results in the specific elimination of many repetitive sequences. This specificity relies on the target-directed scan RNA degradation (TDSD) pathway where scnRNAs matching the active parental macronucleus are eliminated.The manuscript by Shehzada et al. identifies a novel player in Tetrahymena TDSD: SUMO E3 ligase Ema2. The authors show by northen and small RNA-seq that Ema2 is required for TDSD. Furthermore, the paper describes how Ema2 post-translationally modifies the transcription elongation factor Spt6 by SUMOylation and that Ema2 is required to produce long doublestranded scnRNA precursor transcripts from the parental macronucleus, possibly via its modification of Spt6.Major commentsFrom Figure 4C, the authors conclude that "Ema2 is the major SUMO E3 ligase during the midconjugation stages.", yet in Figure 5 show that only Spt6-SUMOylation is affected in Ema2 mutants. These conclusions seem inconsistent and should be reconciled as it is a central point in the paper. E.g. is Spt6 protein abundance based on the MS data supporting that this protein constitutes a major fraction of the (high mol weight) SUMOylated proteins? Of note, the discussion contains a very balanced discussion of this but the current description in the results should be improved.

Some of the proteins detected from both the wild-type and *EMA2* KO conditions were possibly poly-histidine-containing proteins that bound intrinsically to the nickel-NTA beads or proteins unpacifically bound to some of the bead material. Taking these possibilities into account, a control experiment with wild-type cells not expressing His-Smt3 in the same condition is now included in the study and any proteins that were also identified in this experiment with log2 LFQ score above 25 were excluded in the new Figure 5A. We also removed any identified proteins containing more than 6 consecutive histidine residues from the plot. After these filtering processes, it is now clear that Spt6 is the major SUMOylated protein detected in the wild-type (with His-Smt3) condition and the LFQ intensities of other proteins (except Smt3) were ~16 or more hold less than that of Spt6. Together with the fact that the molecular weight range of most of the SUMOylated proteins fits very well to that of SUMOylated Spt6, we are now more confident to conclude that Ema2 is the major SUMO E3 ligase during the mid-conjugation stages and Spt6 is the major target of Ema2. We have modified the corresponding figure and texts to explain this filtering and the outcomes (Page 9, Lines 2-9).

The western blots carried out for the chromatin fraction and presented in Figures 7B, 7C, and 8B have variable levels of histone H3 which serves as a fractionation control, thus indicating some experimental variability. To support the quantitative conclusions, the authors should indicate how many times were these fractionation experiments repeated and should also provide experimental replicate data in the supplements. These data are important to firmly support the quantitative conclusions the authors currently draw from the experiments.

Each of these fractionation experiments was done three times and gave comparative results. The replicate data have been shown in Figures EV5, EV6 and EV8.

Minor commentsPage 3: "Because small RNA-producing loci are also small RNA targets … " It should be specified that this is the case specifically for the studied system as it is not generally the case for small RNA loci. Overall, this third intro paragraph is a bit hard to read and might be improved by first introducing Tetrahymena and its distinctive cellular biology and then moving to the observation that small RNA source and target loci are separated in this ciliate.

We have modified the description to “Because small RNA-producing loci are also small RNA targets in most of the studied small RNA-directed heterochromatin formation processes, it poses a challenge to separately investigate lncRNA transcription for small RNA biogenesis and that for small RNA-dependent recruitment of downstream effectors in these processes.” (Page 3, Lines 24-27). We believe this has improved overall readability of the paragraph.

Figure annotation and readability: The manuscript and figure labels are rich in abbreviation (and sometimes even abbreviations of abbreviations, e.g. na = new MAC = new macronucleus).

We agree that there are many abbreviations in this manuscript but we believe most of them are necessary to keep the text and figures concise. To increase readability, we have spelled out all “abbreviations of abbreviations” when they appear the first time in the text. In fact, “na” was used not as an abbreviation but as a mark in the figures. We have modified the corresponding figure legends to make this point clearer. Also, to make the abbreviation “TDSD” more generalizable, we modified the manuscript to used it as “target-directed small RNA degradation” instead of “target-directed scnRNA degradation”.

Also Figures 4, 5 – the addition of the protein name after α-HA, -GST or -His would make the interpretation of blots easier.

Because anti-GST is detecting both GST alone and GST-Ema2, in Figure 4B, we had indicated the names of the proteins next to the blots. These might be less visible due to the busy arrangement of the panels in the previous manuscript. We have made extra space to make these labeles more visible. For Figure 4C, Figure 5B and Figure 5C, we have followed the reviewer’s suggestion and changed the labels to show the proteins detected.

In Figure 4, it is unclear how the protein quantification was made (leading the "reduced to ~50% in the EMA2 KO" statement). Please clarify.

The total signal intensities of HA-Smt3 in triplicated experiments were analyzed by western blotting and quantified. We now have included the data as a part of Figure 4C in the revised manuscript and explained the quantification procedure in the figure lagend and Materials and Method.

In some places, the current manuscript refers to implicit knowledge that some non-specialists may not take for granted. For example, dsRNA formation is important for scnRNA production, motivating detection using the J2 antibody. Editing for non-expert readability could help reach a broader readership.

In this study, we used the J2 antibody not because dsRNA formation is important for the scnRNA production but because it allows us to cytologically detect lncRNAs in the parental MAC. We have modified the related sentence (Page 10, Lines 17-20) in the revised manuscript to improve readability. We have also added a discussion about single vs double-strand nature of lncRNA in the parental MAC (Page 10, Lines 30-34) as mentioned in our reply for the first comment of Reviewer 1.

Also, on Page 7, bottom, it would be helpful to briefly explain to the reader how SUMOylation works to motivate the conclusion from the Ubc9 interaction.

We have added a brief explanation for the actions of E1 and E2 enzymes in SUMOylation in the revised text (Page 8, Line 6-7).

Referees cross-commentingMy report (rev #2) closely aligns with that of rev #3. While all reports are positive, rev #1 suggests several lines of additional work, such as the characterization of lncRNA expression in the new MAC (major concern 3) and a search for other SUMO E3 ligase (major concern 4). While several interesting ideas are brought up here, I see such added investigations as non-essential for the current paper. I would encourage to focus revision work on the substantiation of the already included experiments.

The lncRNA expression in the new MAC in the *C-KR* mutant has been analyzed and included in Figure EV9. We have included some discussion regarding other SUMO E3 ligases and reserved their functional investigations for our future studies as Reviewer #2 and #3 suggested.

Reviewer #2 (Significance (Required)):SignificanceOverall, the presented work is well-structured, well-executed experimentally and carefully interpreted. The manuscript in most places (see minor comments) is clear and easy to follow for the expected broad readership in the fundamental biology of small RNAs and programmed DNA elimination. The main weakness of the paper is the proposed mechanistic connection from the Ema2 KO phenotype to Spt6 SUMOylation function in TDSD. The authors, however, have a very balanced description of this aspect in the discussion. In addition, there are some important technical questions to address regarding protein quantification by western blotting.The work presented elucidates the crucial role of SUMO E3 ligase Ema2 in the TDSD pathway for scnRNAs in Tetrahymena. This advance is significant as TDSD is the foundation for the specificity of programmed DNA elimination in Tetrahymena and as it is currently not well understood mechanistically.This work will be of interest to a broad readership for two reasons: (i) it advances our understanding of programmed DNA elimination in Tetrahymena, which is a major mechanistic model system for eukaryotic programmed DNA elimination. And (ii) it makes mechanistic connections to small RNA-mediated transcriptional silencing in yeast and fruit flies with possible general implications for these processes across eukaryotes.In sum, the paper presents interesting new findings about small RNA biology and DNA elimination and was a pleasure to read.The reviewers' declared field of expertise: small RNAs, chromatin, transcriptionReviewer #3Reviewer #3 (Evidence, reproducibility and clarity (Required)):This study presents novel data and evidence for a critical involvement of protein SUMOylation in the process of noncoding RNA transcription during the process of conjugation in Tetrahymena. Loss of the critical SUMO E3 Ligase Ema2 leads to a loss of ncRNA transcription in the parental macronucleus, ultimately leading to the lack of scanRNA traget molecules on chromatin, and as a result a loss of heterochromatin formation as well as defective target-dependent small RNA degradation.The paper is very well written, the figures are mostly a treat, the data is well discussed and placed in context, and the claims are supported by robust data. The authors went a long way to nail the relevant target protein of Ema2 and provide on the one side compelling evidence that the transcription elongation factor Spt6 is a bona fide SUMOylation substrate for Ema2. Quite surprisingly, however, a mutant Spt6 construct that shows no sign of SUMOylation in cells does rescue the Spt6 loss of function phenotype. While this puts the relevance of Spt6 SUMOylation in the process slightly into question, the authors provide a compelling discussion as to how SUMOylation still might be essential for proper Spt6 function in stimulating ncRNA transcription. All in all, this is a great paper that reports important data for the ciliate community, for the transcription community, and the larger small RNA community.The following comments hopefully help to further improve the paper. I do not recommend any additional experiments.Introduction: It is not entirely clear why the transcripts of small RNA targets are necessarily noncoding. labelling them as nascent would be sufficient in my opinion

In the described examples of small RNA-directed heterochromatin formation processes in the various eukaryotes in Introduction, the targets of small RNAs are indeed lncRNAs. Therefore, to separately discuss small RNA targets from mRNA, we keep using the term lncRNA for the former. It is unclear whether mRNAs can also be small RNA targets in the *Tetrahymena* DNA elimination process. We have added the following sentence in Introduction (Page 4, Line 30):

“Although mRNAs are transcribed in the parental MAC, it remains unclear if they also can induce TDSD and how mRNAs and pMAC-lncRNAs can be transcribed from overlapping locations.” Nonetheless, because *EMA2* KO did not show detectable defect in the progression of conjugation processes, we believe any essential mRNA transcriptions for these processes occur in the parental MAC in *EMA2* KO (which are now mentioned in Discussion [Page 13, Lines 14-20] for replying to one of Reviewer 1’s suggestions) and thus believe that the defects of *EMA2* KO observed/discussed in this manuscript are due to the loss of lncRNAs. Therefore, we believe using lncRNA to label the RNAs transcribed by Ema2-directed SUMOylation is valid.

The nomenclature of methylated H3K9 might need some adjustment. Consider the abbreviation H3K9me2/3 instead of H3K9me

We followed the suggestion and H3K9me2/3 or H3K9m3 have been used in the revised manuscript.

It would be desirable if the authors could cross reference to the Paramecium field where possible given that this is a second, powerful study system in small RNA-mediated genome elimination.

We have extensively modified Introduction to describe the small RNA-directed genome rearrangement process of *Tetrahymena* and *Paramecium* as much as possible in parallel.

Main text:"The conjugation-specific expression and the localization switch from the parental to the new MAC are reminiscent of the factors involved in DNA elimination (Mochizuki et al., 2002; Coyne et al., 1999; Kataoka & Mochizuki, 2015; Liu et al., 2007; Yao et al., 2007)." please name these other factors here.

We have added “such as the Piwi protein Twi1, which is loaded by scnRNAs, and PRC2 (Mochizuki et al. 2002; Liu et al. 2007; Noto et al. 2010)” at the end of this sentence (Page 6, Line 13).

Figure 5A: what is the author's interpretation of the finding that most identified proteins remain unchanged? are these Ema2 independent SUMOylated proteins or are these background proteins that are not SUMOylated?

As mentioned in our reply to Reviewer 2, some of the proteins detected from both WT and *EMA2* KO were possibly poly-histidine-containing proteins that bound intrinsically to the nickel-NTA beads without His-Smt3 conjugation or proteins unpacifically bound to some of the bead material. Taking these possibilities into account, a control experiment with wild-type cells not expressing His-Smt3 in the same condition has now been included and any proteins that were also identified in this experiment with log2 LFQ score above 25 were excluded in the new Figure 5A. We also removed any proteins containing more than 6 consecutive histidine residues from the plot. After these filtering processes, it is now clear that Spt6 is the major SUMOylated protein detected in the wild-type (with His-Smt3 expression) condition and the LFQ intensities of other proteins (except Smt3) were ~16 or more hold less than that of Spt6. We have modified the corresponding figure and texts (Page 9, Lines 2-9) to explain this filtering procedure and the outcomes. Even after this filtering, many proteins were identified similarly between wild-type and *EMA2* KO conditions. As mentioned in our reply for one of the comments by Reviewer 1, these are most likely Ema2-independent SUMOylated proteins either mediated by another SUMO E3 ligase or by E3-independent SUMOylation. We have added these points in the revised manuscript (Page 8, Lines 22-25).

"However, the cells rescued by HA-SPT6N-KR and HA-SPT6-M-KR showed severe defects in meiotic progression and mating initiation, respectively, making their SUMOylation status during conjugation uninvestigable." Why can't you investigate the SUMOylation capacity of these variants in wildtype cells?

The suggested experiment is probably a valid way to investigate the SUMOylation of HA-Spt6NKR and HA-Spt6-M-KR. However, in such experimental setting, SUMOylation of Spt6 might be blocked not by loss of SUMOylation sites but by competition between the wild-type and the mutant Spt6. Moreover, even if one of them is proved to be unSUMOylatable (we now decided to call it SUMOylation-defective mutant [please see below]), we cannot examine its effect on lncRNA transcription if it has to be co-expressed with the wild-type Spt6. Therefore, we decided not to further examine the SUMOylation of the two mutants.

"Therefore, Spt6-C-KR is an unSUMOylatable Spt6 mutant." How sure can you be about this given the dynamic range of the detection in this experiment?

Whatever the dynamic range is, it is not possible to conclude that there is zero SUMOylation on Spt6-C-KR in the experimental setting we used. So, we have decided to call it a “SUMOylation defective mutant” and modified the corresponding sentence as follows (Page 12, Line 18): “Therefore, Spt6-C-KR represents a SUMOylation-defective Spt6 mutant, exhibiting at least a reduced level of SUMOylation compared to Spt6 in the absence of Ema2 (compare Figure 8B and Figure 5B).”

Figure 1A: label the plot to make it more accessible. Axis labels are missing.

Axis labels and explanations for the stages have been added in the revised Figure 1A.

Figure 3A: can you speculate about the higher molecular weight signal in the northern blot that appears in the later time-points and that seems to be partially dependent on Ema2?

The appearance of these higher molecular weight signals correlates with the presence or absence of lncRNAs detected by the J2 antibody at 4.5 hpm (Figure 6B). However, their presence in *EMA2* KO cells at 6 hpm, the time point before the development of the new MAC, does not fit well to the absence of J2 staining in the parental MAC in *EMA2* KO cells. Therefore, we currently have no clear idea for the identity of the higher molecular weight signals.

Figure 3B: why are the scanRNA levels at 3h already so different between WT and mutant cells?Lane 1 versus lanes 3 and 5?

The following sentence has been added in the revised manuscript (Page 7, Line 20):

“Because TDSD takes place concurrently with the scnRNA production (Schoeberl et al. 2012), the increased abundance of MDS-complementary scnRNAs at 3 hpm in the *EMA2* KO cells compared to the wild-type cells can also be attributed to the necessity of Ema2 in TDSD.”

Figure 5: could you comment on the weak Smt3 signal that remains for Spt6 in the Ema2 KO conditions. Is this due to other SUMO-ligases or is the Ema2 KO not a full loss of function condition?

The following sentence has been added in the revised manuscript (Page 9, Line 31):

“The remaining SUMOylation observed on Spt6 in the absence of Ema2 is likely facilitated by other SUMO E3 ligases and/or E3-independent SUMOylation, as discussed earlier for the other instances of Ema2-independent SUMOylations.”

Figure 6C: are the many arrowheads not confusing? Are they needed?

We have removed most of the arrowheads from the figure and marked only the parental MACs. In addition, we have used the same labeling for all immunofluorescent staining figures.

Figure 8A: the cartoon depicting different colors for the various Lysine residues is not immediately clear to the reader. Try to make this more accessible.

We have modified the drawing to make the markings for the mutated lysine residues more visible in the revised figure.

Referees cross-commentingI agree with the comment from reviewer #2 that additional experiments are not required at this stage. Several constructive points have been raised by all three reviewers that will strengthen this already very mature work.Reviewer #3 (Significance (Required)):This is a very strong experimental study that reports very interesting findings that do go beyond the ciliate community. Spt6 is a major transcription elongation factor and understanding the various functions of this factor by studying in vivo processes is highly important. The paper opens up a new research niche. The findings are very well presented and the discussion does a great job in putting the somewhat surprising results n the non SUMOylatable mutant into context.